# Reduced exposure to extreme precipitation from 0.5 °C less warming in global land monsoon regions

Wenxia Zhang [1,2], Tianjun Zhou [1,2], Liwei Zou [1], Lixia Zhang[1,3] & Xiaolong Chen [1]

The Paris Agreement set a goal to keep global warming well below 2 °C and pursue efforts to limit it to 1.5 °C. Understanding how 0.5 °C less warming reduces impacts and risks is key for climate policies. Here, we show that both areal and population exposures to dangerous extreme precipitation events (e.g., once in 10- and 20-year events) would increase consistently with warming in the populous global land monsoon regions based on Coupled Model Intercomparison Project Phase 5 multimodel projections. The 0.5 °C less warming would reduce areal and population exposures to once-in-20-year extreme precipitation events by 25% (18–41%) and 36% (22–46%), respectively. The avoided impacts are more remarkable for more intense extremes. Among the monsoon subregions, South Africa is the most impacted, followed by South Asia and East Asia. Our results improve the understanding of future vulnerability to, and risk of, climate extremes, which is paramount for mitigation and adaptation activities for the global monsoon region where nearly two-thirds of the world's population lives.

[1] State Key Laboratory of Numerical Modeling for Atmospheric Sciences and Geophysical Fluid Dynamics, Institute of Atmospheric Physics, Chinese Academy of Sciences, Beijing 100029, China. [2] University of Chinese Academy of Sciences, Beijing 100049, China. [3] Collaborative Innovation Center on Forecast and Evaluation of Meteorological Disasters, Nanjing University of Information Science & Technology, Nanjing 210044, China. Correspondence and requests for materials should be addressed to T.Z. (email: zhoutj@lasg.iap.ac.cn)

Faced with the threat of ongoing climate change, the 2015 Paris Agreement proposed an ambition to "hold[ing] the increase in the global average temperature to well below 2 °C above preindustrial levels and pursue[ing] efforts to limit the temperature increase to 1.5 °C, recognizing that this would significantly reduce the risks and impacts of climate change"[1]. Significant effort has since been devoted to understanding the different climatic impacts of the two warming levels, including changes in temperature and precipitation extremes on both global and regional scales[2–7].

Precipitation-related extremes are among the most impact-relevant consequences of a warmer climate, particularly in the global monsoon regions (regions surrounded by magenta lines in Fig. 1; see Methods). The global monsoon is characterized by pronounced annual variation in precipitation and low-level winds, which are originally driven by insolation[8–10]. Sufficient monsoon rainfall and, hence the rich freshwater resources sustain approximately 62% of the world's population (Fig. 1a). Nevertheless, the global land monsoon (GM) region has been overwhelmed by extreme precipitation. The annual maximum accumulated 5-day precipitation (RX5day), a frequently used index of extreme precipitation in flood risk assessments[11], over the monsoon regions is far greater than that over the rest of the land. Climatologically, the RX5day reaches as high as 117 mm averaged over the GM region, as compared to 53 mm for the rest

of the global land, estimated using the gauge-based gridded daily precipitation from the Global Precipitation Climatology Centre[12] (Fig. 1b). Such types of excessive extreme precipitation can cause severe floods and even landslides and debris flows in mountainous areas in the GM domain. Extreme precipitation in the monsoon regions is projected to further intensify with warming[13,14]. Assessments of the changing associated risks, especially the impacts that could be avoided by limiting warming to 1.5 °C compared to 2 °C, are critical for mitigation and adaptation planning.

In this study, we investigate the future changes in extreme precipitation in the GM region based on RX5day, by employing multimodel projections from the Coupled Model Intercomparison Project Phase 5 (CMIP5) (Supplementary Table 1; see Methods; ref. [15]). Climate change risks are typically determined by the hazards, vulnerability and exposure of human society and natural ecosystems[16,17]. Vulnerability is a function of exposure, sensitivity, and adaptive capacity[16–18]. Here, we quantify the changes in exposure to extreme precipitation at different warming levels, focusing on the different impacts at the 1.5 and 2 °C warmer worlds in particular. We show that both the area and population exposed to dangerous precipitation extremes would increase consistently with warming. Realizing the 1.5 °C low warming target would robustly reduce the areal and population exposures to dangerous extremes for the populous GM region, compared to a warming of 2 °C. The avoided impacts are more remarkable for more intense extremes. Such information is fundamental for understanding future vulnerability and for developing mitigation and adaptation strategies.

## Results

**Response of extreme precipitation to warming.** Over the GM region, the long-term change in RX5day is dominated by global warming and features spatially consistent increases, as shown by the leading mode of the Empirical Orthogonal Function derived from historical simulations and projections under the Representative Concentration Pathway (RCP) 8.5 between 1860 and 2100 from the CMIP5 multimodels (Supplementary Fig. 1). Up to approximately 40% of the total variance is explained by the global warming mode for the individual models.

The RX5day averaged over the GM region responds approximately linearly to the global temperature increase at a rate of 5.17% $K^{-1}$, with a 25th–75th percentile range of 4.14–5.75% $K^{-1}$ (Fig. 2a and Supplementary Fig. 2; see Methods). Such a response is generally consistent with (although slightly lower than) that expected from the Clausius–Clapeyron equation. The thermodynamic arguments suggest an increase in heavy rainfall intensity at a rate similar to the moisture increase of 7% $K^{-1}$, as heavy precipitation is largely driven by moisture convergence[19]. The slightly weaker response of the RX5day in simulations compared to the thermodynamic arguments implies a potential offset from the dynamic changes overall. Changes in dynamic circulation can substantially affect extreme precipitation[14,20–22]. Underlying mechanisms include the weakening of the large-scale monsoon circulation[13,23–26], which results from the stabilization of the tropical atmosphere associated with global warming[27,28], the modulation of regional monsoon circulation by land–sea thermal contrast changes[29,30], the gradient of sea surface temperature warming patterns[31], and changes in synoptic scale circulations, such as monsoon depressions[22,30,32] and tropical cyclones[33].

The spatial pattern of extreme precipitation response is not homogeneous mainly due to the dynamic contribution[22]. Specifically, extreme precipitation in the South and East Asian monsoon regions is the most sensitive to warming, with a response rate of 9.67% $K^{-1}$ (7.00–10.41% $K^{-1}$) and 6.40% $K^{-1}$

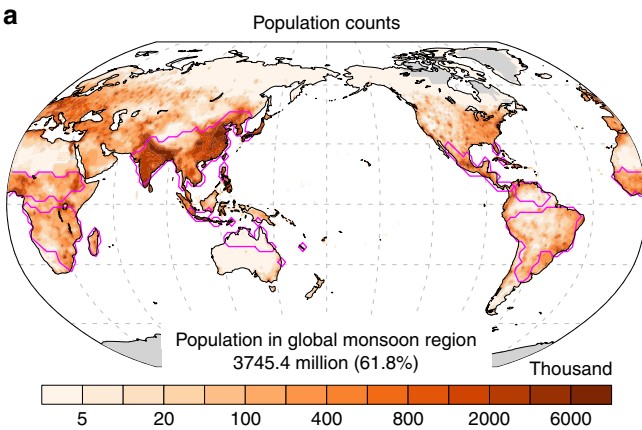

**a**

Population counts

Population in global monsoon region
3745.4 million (61.8%)

Thousand

| 5 | 20 | 100 | 400 | 800 | 2000 | 6000 |

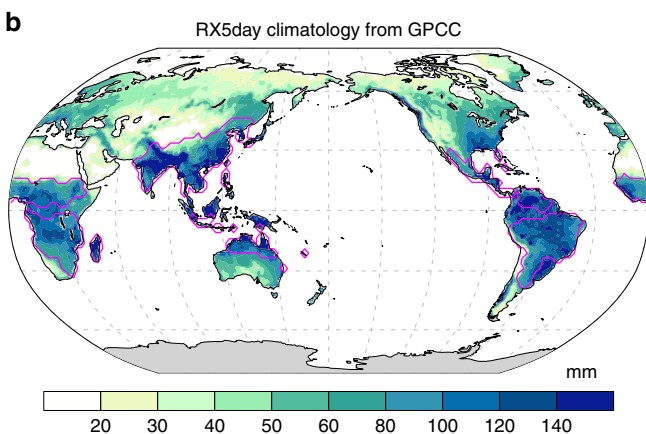

**b**

RX5day climatology from GPCC

mm

| 20 | 30 | 40 | 50 | 60 | 80 | 100 | 120 | 140 |

**Fig. 1** High impact of extreme precipitation on the global land monsoon region. **a** Global population in 2000 from the Gridded Population of the World[35] (GPW2000; thousand per 1° × 1° grid). **b** Climatological maximum accumulated 5-day precipitation (RX5day) from GPCC[12] in 1988–2013. Magenta lines denote the GM region (see Methods)

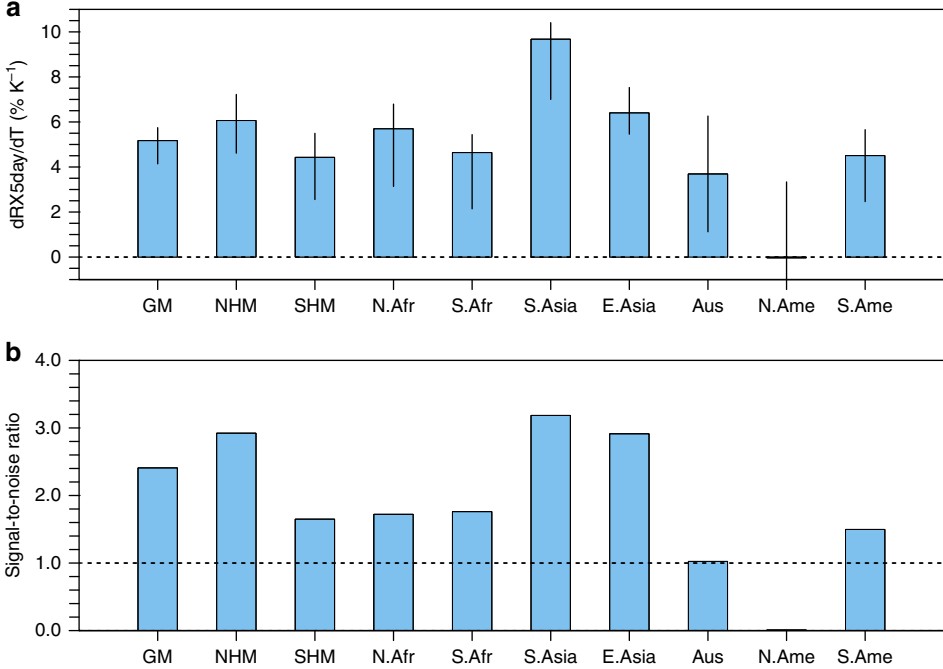

**Fig. 2** Response of RX5day to global warming. **a** Multimodel ensemble medians (histograms) and interquartile ranges (error bars) of the RX5day response to global mean surface air temperature changes over the GM and individual monsoon regions (see Methods). NHM: northern hemispheric monsoon; SHM: southern hemispheric monsoon; N.Afr: North African monsoon; S.Afr: South African monsoon; S.Asia: South Asian monsoon; E.Asia: East Asian monsoon; Aus: Australian monsoon; N.Ame: North American monsoon; S.Ame: South American monsoon. **b** Signal-to-noise ratios of RX5day responses. Signal refers to multimodel median responses, and noise refers to intermodel standard deviations

(5.46–7.53% K$^{-1}$), respectively (Fig. 2a). These responses are robust against model spread, as shown by their high signal-to-noise ratios (SNRs; Fig. 2b; see Methods). In contrast, the North American and Australian monsoon regions exhibit moderate responses with low SNRs, which are partly related to their relatively smaller areal coverage and competing roles of thermodynamics and dynamics[22].

In the 1.5 and 2 °C warmer worlds, consistent increases in RX5day are projected throughout the GM region, except over the North American monsoon region (Fig. 3). Robust increases in RX5day are projected to mostly affect the Asian and African monsoon regions with the half a degree additional warming due to the large sensitivity of extreme precipitation to global warming in these regions (Fig. 3c). Note these regions have dense populations (Fig. 1a).

**Exposure to dangerous extreme precipitation events.** In terms of social impacts, extreme events that deviate substantially from their climatologies can result in the greatest losses (for a certain region, regardless of the changes in societal factors such as vulnerability), because they are beyond the tolerable ranges of ecological and human systems and infrastructures. The response of extreme precipitation to global warming is twofold: mean state and variability. CMIP5 models show increases in both the mean state and the variability of extreme precipitation with warming (Supplementary Fig. 3), consistent with previous studies[34]. Increases in both the mean state and variability would increase the frequency of intense extreme events that could be dangerous in terms of social impacts. Here, we define dangerous extreme events as those exceeding the 10- and 20-year return values from the 1950–2005 baseline, which lie in the upper tail of the extreme value distributions (Supplementary Fig. 4; see Methods). The two thresholds represent different levels of dangerous. We then estimate the areal and population exposures to these dangerous

extremes for different warming levels for the integrated GM region and regional monsoon domains, respectively (see Methods). The population exposure is estimated based on the population distribution fixed at the year 2000 (Gridded Population of the World, GPW2000; ref. [35]) and that projected under different socioeconomic development scenarios of Shared Socioeconomic Pathways[36] (SSPs).

The evolution of exposure with warming levels indicates the speed at which the human system will be hit by these dangerous extremes. The land area exposed to these events increases consistently with warming (Fig. 4a). For the GM region as a whole, for RX5day events that exceed the baseline 10-year return values, the area of exposure increases from the present-day (1986–2005) level of 9.47% (9.13–9.89%) to 12.42% (10.46–13.53%) and 14.01% (12.55–15.98%) for the 1.5 and 2 °C warming levels, respectively. Similarly, the area exposed to RX5day events that exceed the baseline 20-year return values increases from 4.34% (4.23–4.58%) for the present day to 6.25% (5.28–7.21%) and 7.45% (6.76–8.57%) for warming of 1.5 and 2 °C, respectively. Correspondingly, the population exposure also increases with continuous warming under the fixed population case (GPW2000) and all projected population scenarios from the SSPs (Fig. 4a). The increases in fractional population exposure are comparable to those in the land fraction.

We further compare the simulated increasing exposure with warming levels to that linearly extrapolated from the preindustrial (0 °C) and 1.5 °C warming levels (dashed gray lines in Fig. 4a). Whereas the increase in exposure is quasilinear below the 2 °C warming level, nonlinear increases emerge at higher warming levels (higher than 2 °C), implying that excessively amplified impacts from dangerous extremes could be induced.

**Avoided impacts by the half a degree less warming.** What are the avoided impacts of the 1.5 °C warming compared to the 2 °C

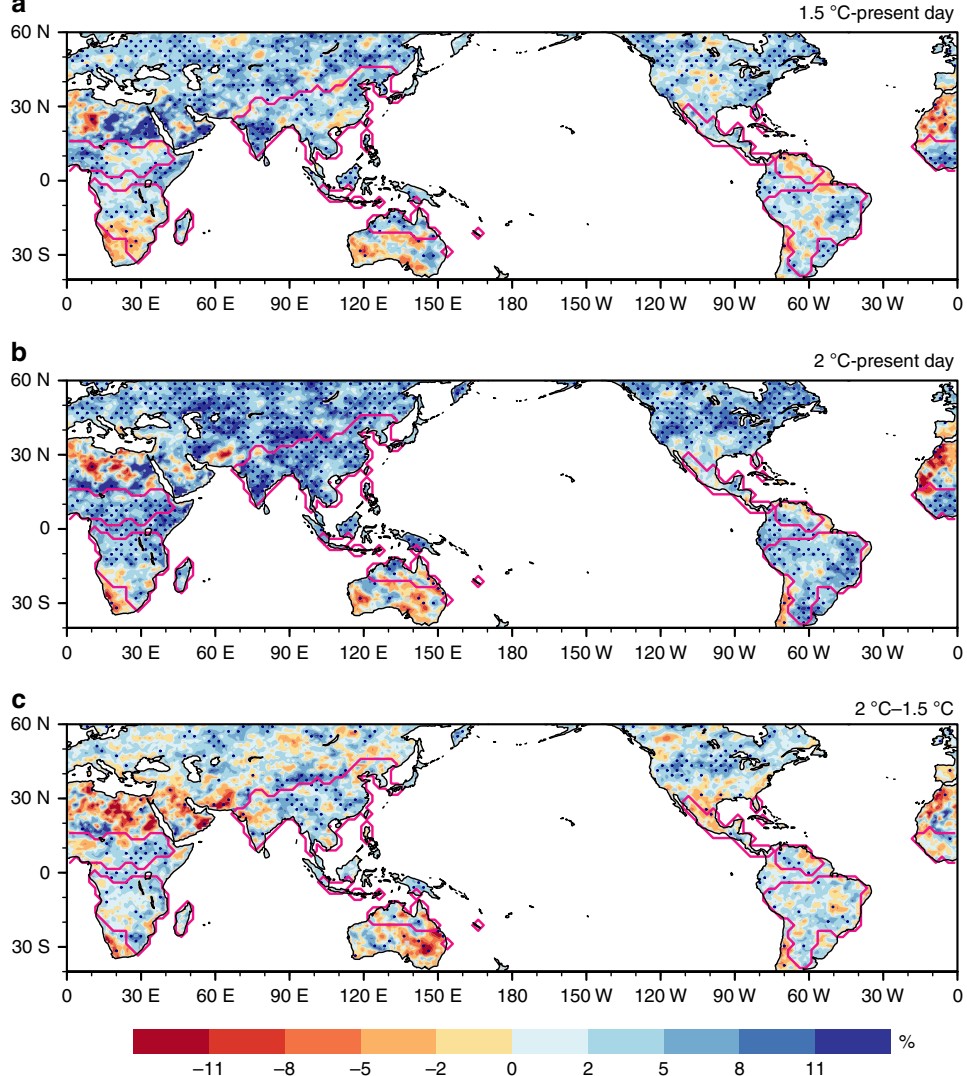

**Fig. 3** Changes in RX5day at different warming levels. **a–c** Multimodel ensemble median changes in RX5day (% relative to the 1986–2005 present-day level) from the present day to 1.5 °C (**a**), the present day to 2 °C (**b**), and 1.5 °C to 2 °C warming (**c**). Stippling denotes where at least 2/3 of the models agree on the sign of the changes. Magenta lines denote the GM region (see Methods)

warming? We quantify this as the difference in impacts (i.e., exposure) between the two warming levels, expressed in percentages with respect to that in the present day (see Methods). If warming is limited to the 1.5 °C level instead of 2 °C, the GM region is projected to benefit from a robustly reduced exposure to dangerous extremes with high model agreement in terms of both area and population (Fig. 4b). Over the integrated GM region, for RX5day events that exceed the baseline 10-year return value, the areal and population (GPW2000) exposures will increase to 120% (112–143%) and 124% (108–136%) over the present-day level for the 1.5 °C warming, respectively, as compared to 146% (130-164%) and 149% (138-167%) for the 2 °C warming, respectively. Thus, the avoided impacts are estimated to be 19% (13–31%) and 27% (13–36%), respectively, for areal and population exposures to the baseline 10-year return value exceedances due to the half a degree less warming. It is worth noting that the avoided impacts are more remarkable for more intense extremes. Thus, for the baseline 20-year return value exceedances, the areal and population exposures that could be reduced by the half a degree less warming amount to 25% (18–41%) and 36% (22–46%), respectively.

Nearly all regional monsoon domains would see such robustly avoided impacts, although the magnitudes would differ (Fig. 5). Hotspots where the avoided impacts are the most prominent are seen in the South African (with multimodel median estimates for avoided areal and population exposures to extremes that exceed the baseline 20-year return values of 44% and 53%, respectively, over the present-day level), South Asian (40 and 40%), and East Asian (35 and 29%) monsoon regions. Here, we only show the population exposure estimated from the fixed population at year 2000 (i.e., the GPW2000 case), while the population exposures based on projections from SSPs are qualitatively similar (Supplementary Fig. 5). Additionally, we note that we only consider the fractional population exposure here. If the absolute population growth is considered, the avoided impacts will be larger.

## Discussion

In summary, we show evidence that limiting global warming to 1.5 instead of 2 °C would robustly reduce areal and population exposures to dangerous extreme precipitation events, i.e., those

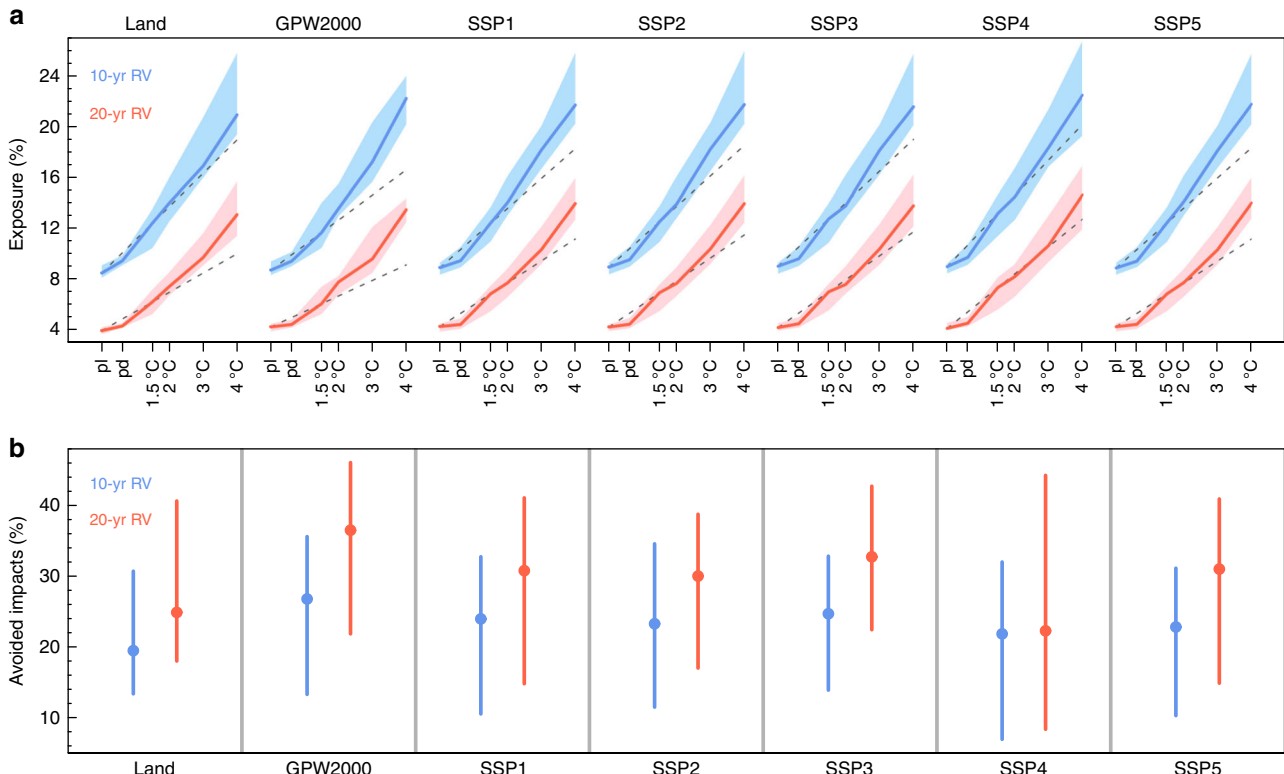

**Fig. 4** Areal and population exposures to dangerous RX5day events over the GM region. **a** Fraction of land area and population experiencing RX5day events that exceed baseline 10- and 20-year return values (RVs) at the preindustrial (pl, 1861–1890, 0 °C), present-day (pd, 1986–2005), 1.5, and 2–4 °C warming levels, over the GM region. Population in 2000 (GPW2000) and under SSPs 1–5 in 2100 are used to estimate population exposure. The multimodel medians (solid lines) and interquartile ranges (shadings) are shown. The abscissa in (**a**) is proportional to the warming magnitudes, where a warming of 0.61 °C is set for 1986–2005 (ref. [45]). The dashed gray lines denote the linear extrapolation from the preindustrial (0 °C) and 1.5 °C warming levels. **b** Areal and population exposures reduced by the 1.5 warming compared to 2 °C warming for RX5day events that exceed the baseline 10- and 20-year return values (see Methods). Circles and bars denote multimodel medians and interquartile ranges, respectively. Where more (less) than 2/3 of the models indicate reduced exposure by the 0.5 °C less warming are indicated by solid (open) circles

exceeding the 10- and 20-year return values for the baseline. The avoided impacts are more remarkable for more intense extremes. Over the GM region, an estimated 25% (18–41%) and 36% (22–46%) of area and population, respectively, could be relieved from the baseline once-in-20-year events over the present-day level, if global warming were limited to 1.5 instead of 2 °C. Among the regional monsoon domains, the primary hotspots of the South African and South Asian monsoon regions, which are already among the most vulnerable regions around the world to adverse impacts of climate change[37], would benefit most from the half a degree less warming with high model consistency, in terms of lower exposure to dangerous precipitation extremes. Such climate change inequity between greenhouse gas emitters and those burden the negative impacts of climate change (including the African countries) has received growing attention[38,39].

Above conclusions do not rely on the definition of dangerous events. Alternatively, if dangerous extreme events are defined as those exceeding 1–2$\sigma$ (interannual standard deviation in the baseline) from the baseline climatology, the exposure and avoided impacts are quantitatively comparable with those based on return values (Supplementary Figs. 6 and 7; see Methods). In addition, analysis of the future emission scenario of the RCP4.5 yields similar results as the RCP8.5 (Supplementary Figs. 8 and 9). The independence of metrics, future emission scenarios, and population scenarios confirms the robustness of the results, thus adding fidelity to the conclusion that humanity would benefit

from the half a degree lower warming target in the densely populated GM region.

In addition to the impact-relevant extremes, changes in the water cycle, particularly the availability of water, which is the life blood of people, in the 1.5 and 2 °C warming futures deserve further investigation[40].

## Methods

**Models**. Historical simulations and projections under the RCP8.5 and RCP4.5 from 21 models in the CMIP5 archive[15] (Supplementary Table 1) are analyzed. The models are selected based on (1) daily precipitation data availability, (2) a 2 °C warming occurs before the year 2100 under both the RCP4.5 and RCP8.5, and (3) the difference in the timing of the 1.5 and 2 °C warming is no less than 9 years to avoid overlap when 9-year time windows are used to represent respective conditions. Excluding models that produce this transition relatively quickly (i.e., FGOALS-s2 and CSIRO-Mk3-6-0, in which the difference of timing of the 1.5 and 2 °C warming is less than 9 years) does not affect the conclusion. All calculations are conducted on native grids in the models, except when spatial patterns are shown.

**Definition of the global monsoon region**. The global monsoon region is defined as the area where the local "summer minus winter" precipitation rate exceeds 2.0 mm/day and the local summer precipitation exceeds 55% of the annual total[9]. Here, local summer refers to May through September for the northern hemisphere and November through March for the southern hemisphere. In this study, we define the global monsoon region based on the 1979–2010 climatological pre-cipitation from the Global Precipitation Climatology Project[41], and we only con-sider land monsoon regions.

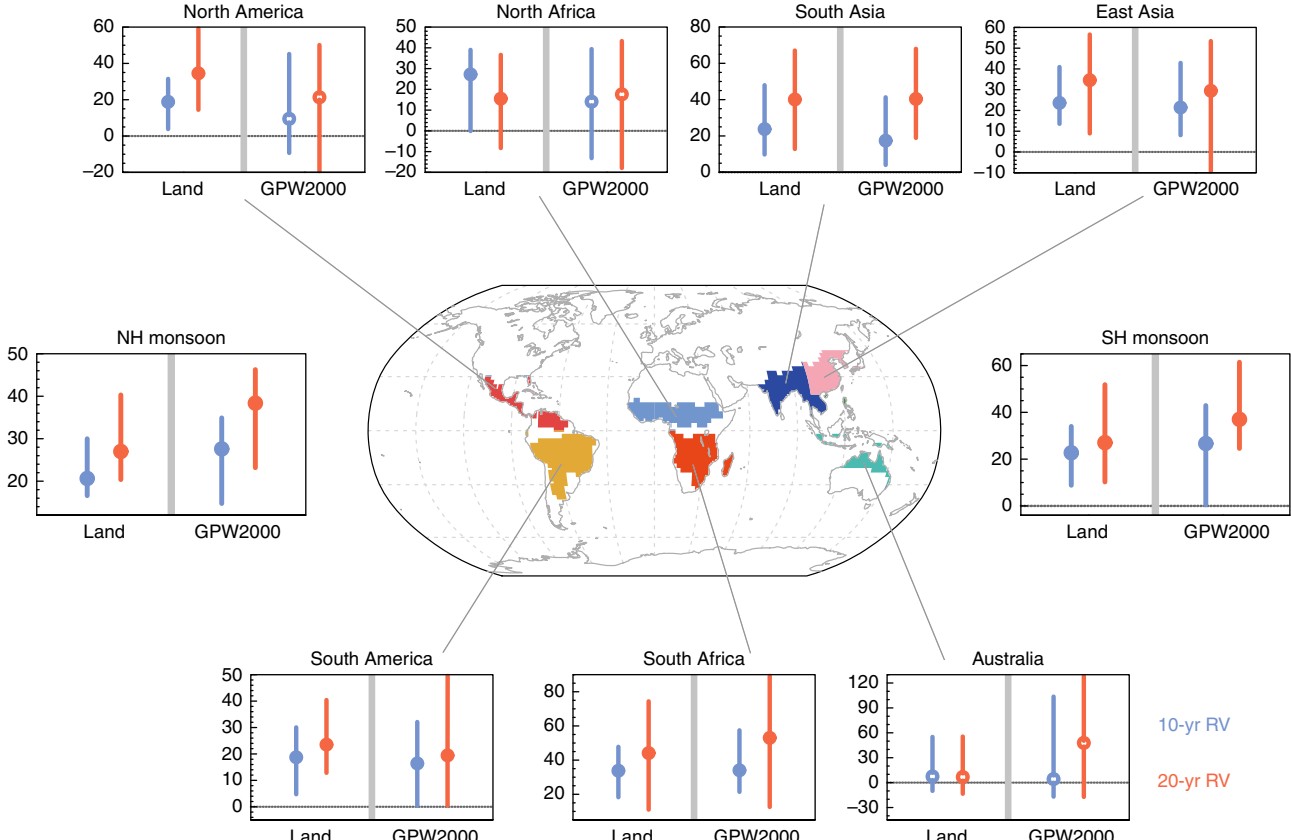

**Fig. 5** Avoided impacts by the half a degree less warming over monsoon subregions. Areal and population exposures (%) reduced by the 1.5 °C warming compared to 2 °C warming for RX5day events that exceed the baseline 10- and 20-year return values (RVs; see Methods). Circles and bars denote multimodel medians and interquartile ranges, respectively. Where more (less) than two-thirds of the models indicate reduced exposure by the 0.5 °C less warming are indicated by solid (open) circles. The population exposures based on population in the year 2000 (GPW2000) are shown. Regional monsoon domains are shown in colors on the map

**Definitions of the present day, 1.5 and 2 °C warmer worlds**. The changes and impacts of the warming scenarios are compared to the present-day conditions, which are defined from 1986 to 2005. The timings of the 1.5 and 2 °C warming scenarios above the preindustrial levels (1861–1890) are determined using the 9-year running average global mean surface air temperature separately for each model. The 1.5 and 2 °C warming periods are then aggregated from the 9-year windows that are centered on the years when respective warming levels are exceeded. Climate changes in the 1.5 or 2 °C warming scenarios are first calculated for each model separately to derive multimodel ensembles. We test whether the multimodel ensemble medians of the 9-year segments are affected by the remaining decadal oscillations. This test is done by comparing the changes in extremes between different warming levels, which are calculated from the original time series and a filtered time series by removing the 10–70-year low-frequency oscillations. The results show that the difference in the multimodel ensemble medians (i.e., signals) is small (Supplementary Fig. 10).

**Response of extreme precipitation to global warming**. To derive the response of extreme precipitation to global warming, projections of extremes and global mean near-surface air temperature are averaged over decadal periods starting in 2006 and overlapped by 5 years (i.e., 2006–2015, 2011–2020, up to 2091–2100). A linear regression between them is referred to as the response rate for each model, separately.

**Signal-to-noise ratio**. The ratio of the multimodel median responses in extreme precipitation to intermodel standard deviations is used as a measure of the SNR[42]. SNRs > 1.0 indicate robust changes compared to the model uncertainty.

**Variability of extreme precipitation**. To calculate the time-dependent inter-annual standard deviation, a local detrending with an 11-year running mean is first applied (i.e., removing the 11-year running mean from the original time series) to derive anomalies for each grid box[43]. Then, the standard deviation is calculated

over running 20-year segments. The standard deviation over the GM region is then derived with area weighting.

**Definition of dangerous extreme events**. The dangerous extreme events in this study refer to those that lie in the upper tail of the extreme value distributions. To objectively define the threshold of dangerous, we employ the 10- and 20-year return values from the 1950–2005 base period to represent different levels of dangerous. This is a period with relatively good global coverage of observations to constrain the models. To estimate the return values for RX5day, a generalized extreme value (GEV) distribution is first fitted to the RX5day in 1950–2005 on the native grids of each model using the method of maximum likelihood[44]. Based on ref. [34], the GEV parameter estimates for extreme precipitation are smoothed spatially, considering the noise in change patterns in extreme precipitation stem from sampling. This is done by smoothing the estimated GEV parameters at each grid point by its eight surrounding neighbors. Then, return values are obtained by inverting the fitted GEV distributions derived from the smoothed parameters. The 10- and 20-year return values from the baseline are derived on the native grid points for each model. The GEV distribution is fitted using the NCL (NCAR Command Language) function "extval_mlegev" (http://www.ncl.ucar.edu/Document/Functions/Built-in/extval_mlegev.shtml).

**Areal and population exposures to dangerous extreme events**. The area (population) that experiences RX5day events exceeding the threshold for danger-ous is aggregated spatially to represent the total area (population) exposed. Frac-tional exposure is calculated with respect to the total area or population over the GM region or monsoon subregions. Population exposures are estimated based on populations from the year 2000 (GPW2000) and under different SSPs to represent all possible future socioeconomic development scenarios. The population expo-sures quantified in the text are based on the GPW2000 case, unless otherwise specified. Generally, fractional population exposure under different SSPs and periods yields the same qualitative results.

**Avoided Impacts.** The impacts in terms of exposure induced by warming are quantified against the 1986–2005 present-day level:

$$\text{Impacts}(K) = \frac{\text{EXP}_K - \text{EXP}_{\text{presentday}}}{\text{EXP}_{\text{presentday}}} \qquad (1)$$

where EXP stands for the exposure and the subscript $K$ indicates the 1.5 or 2 °C warming level. Thus, the impacts avoided by the 0.5 °C less warming are derived as the difference between the impacts at the two levels.

**Sensitivity to definitions of dangerous events.** To confirm the robustness of the conclusion, we also employ another alternative metric to determine the threshold of dangerous, i.e., the $\sigma$-exceedance metric based on normalization. Here, $\sigma$ refers to the interannual standard deviation in the 1950–2005 baseline. The original time series is normalized with the base period of 1950–2005 by removing the mean and then dividing by the standard deviation ($\sigma$). RX5day events that exceed $1.00\sigma$, $1.30\sigma$, $1.65\sigma$, and $2.00\sigma$ are thus identified as dangerous extremes, which correspond to approximately 6-, 10-, 20-, and 44-year return values, respectively, for a Gaussian distribution. We note that normalization is generally used for data with an approximate Gaussian distribution, our use of $\sigma$-exceedances for the skewed extreme value distribution here only serves as a reference threshold to define intense extremes.

We compare the key results of this study between the two metrics, i.e., the return value exceedance and $\sigma$-exceedance (cf. Figs. 4 and 5 and Supplementary Figs. 6 and 7). The comparison shows that the areal and population exposures to dangerous extremes, as well as the impacts avoided by the half a degree less warming are quantitatively comparable between the two metrics. For example, the reduced area of exposure to baseline 20-year return value exceedances (estimated from GEV distributions) is 24.9% (18.0–40.6%), while that to $1.65\sigma$-exceedances (corresponding to the 20-year return value for a Gaussian distribution) is 22.1% (13.3–37.9%). The consistency between the two metrics confirms the robustness of the conclusion.

**Code availability.** The data in this study were analyzed and the figures were created with NCAR Command Language (NCL; ref. [46]). All relevant codes used in this work are available, upon request, from the corresponding author T. Z.

**Data availability.** Population distributions in the year 2000 are from the Gridded Population of the World, version 3 (GPWv3, ref. [35], http://sedac.ciesin.columbia.edu/data/set/gpw-v3-population-count). Future population distributions under different SSPs[36] between 2010 and 2100 are from https://www2.cgd.ucar.edu/sections/tss/iam/spatial-population-scenarios. Daily precipitation data from the Global Precipitation Climatology Centre[12] (GPCC) is from http://gpcc.dwd.de/.

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

## Acknowledgments

This work is supported by National Natural Science Foundation of China under Grant nos. 41330423 and 41420104006, and International Partnership Program of Chinese Academy of Sciences under Grant no. 134111KYSB20160031. L. Zhang is supported by Program of International S&T Cooperation under Grant no. 2016YFE0102400.

## Author contributions

T.Z. designed the research, provided comments, and revised the manuscript. W.Z. performed the analysis and drafted the manuscript. L. Zou, L. Zhang, and X.C. helped to organize and revise the manuscript. All of the co-authors contributed to scientific interpretations and helped to improve the manuscript.

## Additional information

**Competing interests:** The authors declare no competing interests.

