## [Peer Review File · Nature Communications]

Editorial Note: Parts of this peer review file have been redacted as indicated to remove third-party material where no permission to publish could be obtained.

Reviewers' comments:

Reviewer #1 (Remarks to the Author):

Review for "Extra 0.5°C warming increases exposure to floods and droughts in global land monsoon regions" by Zhang Wenxia et al.

Summary:

In the present manuscript, the authors quantify differences in precipitation characteristics focusing on the extreme tails across the world's monsoon regions for present-day climate, and hypothetical 1.5°C and 2°C worlds, including the impact of the half degree difference between the latter on precip. characteristics. Subsequently, the authors go one step further towards quantifying "risk" by analysing changes in precipitation characteristics jointly with population scenarios.

I fully agree with the authors that understanding changes in precipitation characteristics, in particular with respect to the extreme tails of the distribution, is crucial; and a paper that carefully discusses and explains differences between a pre-industrial, and 1.5°C / 2°C worlds (and potentially higher-degree worlds), focusing on (extreme) precipitation induced risks, including a joint discussion/analysis with changes in exposure and/or vulnerability in the world's monsoon region could be an important contribution. Unfortunately, the present paper has little to offer in this regard due to several shortcomings. This includes an unfocused discussion iterating some well-known results and subjective choices/statements, several conceptual issues related to the metrics used and how they are interpreted (i.e. drought vs. precipitation deficit metrics, and flood vs. high precipitation; but also the use of the concept of exposure and vulnerability as determinants of risk, see comments below), and finally issues of technical correctness of the paper related to deriving estimates based on a baseline climatology that could be prone to a potential overestimation of extremes outside the reference period, which requires some testing (please see detailed comments below). Due to these issues, I do not believe that the present manuscript can be considered for publication in the journal "Nature Communications".

Major comments:

(1) Unfocused discussion in many places; lack of discussion of limitations and methodological description

While (again) the topic addressed by the authors is interesting, unfortunately the discussion of results is in many places unfocused, re-iterates several known results, and lacks explanations elsewhere (in particular as the authors present many results). Please find a few examples below:

- The paper relies on an extreme metric that the authors term 'unusual' or 'dangerous' wet extremes, which is defined as "50% greater intensities relative to the 1986-2005 baseline" (l. 20-21) in the Rx5day metric (the reader understands this only from the caption of Fig. 4; and I guess the authors mean "50% greater intensities relative to the Rx5day mean of the 1986-2005 baseline", but this is not indicated). However, it remains entirely elusive as to why the authors choose this metric; or why this metric might be important for impacts such as floods. While, clearly, 50% more than Rx5day (mean) precipitation seems like a large amount, this might differ substantially across regions - i.e. there might be large gradients in the variability of Rx5day across the GM regions. This means that while there might be some regions in which a 50% exceedance of the baseline is a 100-year event, in other regions a 50% exceedance might happen every other year and might thus not really be extreme (e.g. Hansen et al. 2012 discuss temperature extremes relative to their variability). Hence, I believe it would be crucial to discuss (a) why is a 50% exceedance relevant or meaningful?, and (b) what does 50% exceedance mean in terms of the variability of extremes? Please other comments regarding the methodological approach further below.

- The Abstract states that "Both the mean state and the variability of the extremes will intensify"

(l. 15-16) as one of the major findings of the paper and shown also in Fig. 2d. However, this issues is left almost entirely out of the discussion in the main manuscript text. I believe a potential reader would like to understand if there are physical reasons for this kind of change, or whether this is just a statistical findings because mean and variance of extreme value distribution (unlike in a Gaussian distribution, for instance) are not independent? Furthermore, it is not explained how the variability estimates are calculated.

- l. 120-126, the authors describe an extreme flooding event that occurred in India in 2013, which had large negative impacts. However, first, this discussion falls out of the blue as the event is arbitrarily chosen, and does not in any way that it obvious to the reader connect to the more large-scale analysis conducted in the remainder of the paper. Second, and this issue follows up on the Rx5day variability estimates (see my comment above), the authors state: "Thus, we emphasize that future changes in the variability of extremes are as important as the changes in the mean state, as an increase in either can induce more unusual and damaging extreme events." (l. 126-129) While this statement seems broadly plausible (but is not based on an analysis), it remains entirely dubious to me why the authors conclude this from a single case study, or in which other way it might be linked to this discussion.

- l. 86-95 The authors devote quite a long discussion to the relationship between RX5day and R90ptot, R95ptot, and R99ptot - I wonder what is the purpose if this discussion, triggered with "Here we show" in l.86; but not at all mentioned in the Abstract or the objectives of the paper? How does this discussion help to addressing the objectives of the paper? (It is not surprising that these metrics are correlated. Moreover, the correlations are clearly non-linear (Fig. 2c) - hence, fitting a linear regression lines is clearly inappropriate...)

- l. 91-93 It has been shown many times in the literature that GCM simulations under climate change tend to produce increases in precipitation extremes, but decreases in the lower part of the distribution. Of course, it is ok to state this again (l. 91-93), or to present this again (Fig. 2a), but then appropriate original literature should be cited (for instance see Allen and Ingram 2002).

- l. 81: The authors broadly refer to the Clausius-Clapeyron relationship and "dynamic" changes as causes for these observed changes in precipitation extremes - which is certainly correct but seems overly simplistic and rather a standard explanation. Since the authors are focusing on the GM regions, I believe a reader would be interested to understand if there are mechanisms that could change the dynamics in a certain way in the GM regions? I do not think a detailed analysis would be needed, but a few sentences more about the underlying reasons of dynamical changes, and what is specific to global GM regions, would be clearly necessary.

The issues discussed above lead to a paper that does not appear to be very focused towards achieving certain objectives, and as a consequence does not seem to present or explain novel findings beyond that extremes in the GM regions are intensifying in global model simulations.

(2) Conceptual issues and methodological limitations need to be discussed:

(a) "Drought" vs. CDDs and Extreme Precipitation vs. "Flood"

The title of the paper is "Extra 0.5°C warming increases exposure to floods and droughts in global land monsoon regions". The authors acknowledge briefly in l. 66-67 that their analysis only deals with the precipitation-related components of flood and drought risks; however, the broader implications of this limitation for the analysis would need to be discussed, in particular if the paper is intended for a broad readership journal. For example, even disregarding the impact on people (vulnerability and exposure; see comment below), the physical origins of droughts are by far not only determined by CCD. See for instance Sheffield et al. 2012; which explains in large detail that there are various feedback mechanisms at play (briefly mentioned by the authors, l. 57-61), but not all of these need to be negative in a warming climate (Seneviratne et al. 2012). Hence, based on the analysis conducted, it is inappropriate to claim that droughts are increasing across global monsoon regions. Partly, the problem might stem from the fact that the authors aim to cover droughts and floods in a single paper, which leaves relatively little room to give an indication of the drivers, let alone broader implications of changes in either phenomenon (in particular such a complex phenomenon such as drought).

(b) Eventually, the authors address "climate risk" related to "droughts" and "floods" (l. 35-38) .

Risk is typically determined as a function of hazard, vulnerability, and exposure (IPCC, 2012); so I am wondering why the authors only mention hazard and vulnerability in their discussion in l. 35-38; but then address hazard and exposure in their analysis (Fig. 3+4). On another note, while I agree that understanding the dynamics in people exposed to any kind of climate extreme is important in the future (which the authors address towards the end of the paper, e.g. in Fig. 4), it seems unclear to me why the metric "Change in hazard" x "Population count" as addressed in Fig. 3 would be particularly meaningful (-> would big changes in extremes in a sparsely populated place be not important to address?) Please explain/discuss (a) why this metric is relevant, (b) its limitations, and (c) why the relationship between hazard and exposure would be simply multiplicative (and not, for instance highly non-linear towards big population changes and/or climate changes).

(3) Technical Correctness

Extreme indicators derived from percentile estimators of a baseline climatological period are known to be sensitive to an overestimation of extremes in the period outside of the baseline period (see e.g. Zhang et al., 2005), because a 20-year or 30-year baseline period is often too short to derive reliable estimates of extreme statistics. This is also the case for other metrics that are estimated from a climatological baseline (see e.g. Sippel et al. 2017) if aggregated across space, such as wet extremes "with 50% greater intensities relative to the 1986-2005 baseline". Percentile-based indicators are used extensively in the paper (Fig. 2a-c, for instance) as well as the "50% metric" (Fig. 4 and highlight statements in the Abstract), but there is no indication whether the authors have corrected for this potential artefact, or whether the authors have tested if this plays a role here. Although I am not sure whether these potential artefacts are relevant here (they might be small; but can be severe sometimes), I believe it would be crucial to test for these issues (in particular because the 50% metric seems quite arbitrary as discussed above).

Minor comments:

l. 44: "nearly two thirds" -> why not state directly that it is 61.8%?

l. 72: How was the model selection done? Based on an ensemble of opportunity?

l. 104: Are there some models in the multi-model ensemble that show lower or higher sensitivities? Would it be possible to derive across-model confidence intervals?

l. 121-122: you state "history has told that extreme events that deviate largely from the climatology can be the most destructive". In the introduction, and scattered throughout the paper, you state that exposure and vulnerability are also crucial contributors to the impacts of climate extremes. Why is this not so here?

l. 134-135: "50% greater intensities relative to the baseline climatology": Please clarify which metric (I am assuming Rx5day) and whether you mean the mean baseline climatology.

l. 136-138: According to standard literature in the field (e.g. IPCC 2012), vulnerability and exposure are distinct features: Please clarify the terminology.

References:

- Hansen, J., Sato, M., & Ruedy, R. (2012). Perception of climate change. *Proceedings of the National Academy of Sciences*, 109(37), E2415-E2423.
- IPCC 2012. *Managing the risks of extreme events and disasters to advance climate change adaptation: special report of the intergovernmental panel on climate change*. Cambridge University Press.
- Sheffield, J., Wood, E. F., & Roderick, M. L. (2012). Little change in global drought over the past 60 years. *Nature*, 491(7424), 435-438.
- Seneviratne, S. I., Nicholls, N., Easterling, D., Goodess, C. M., Kanae, S., Kossin, J., ... & Reichstein, M. (2012). Changes in climate extremes and their impacts on the natural physical environment. (Chapter 3 in IPCC 2012).
- Seneviratne, S. I. (2012). Climate science: Historical drought trends revisited. *Nature*, 491(7424), 338-339.
- Sippel, S., Zscheischler, J., Heimann, M., Lange, H., Mahecha, M. D., van Oldenborgh, G. J., ... &

Reichstein, M. (2017) Have precipitation extremes and annual totals been increasing in the world's dry regions over the last 60 years? *Hydrology and Earth System Sciences*, 21, 441-458.
Zhang, X. B., G. Hegerl, F. W. Zwiers, and J. Kenyon (2005), Avoiding inhomogeneity in percentile-based indices of temperature extremes, *J. Clim.*, 18(11), 1641-1651

Reviewer #2 (Remarks to the Author):

This is a very interesting and timely work and may serve as a very important document for policy making. In my opinion, it is worth publishing such an article. However, I have couple of points, which the authors may address:

1. There is significant uncertainty across models, and hence we get a box plot of changes derived from the models. Under such an uncertain scenario, statistical significance plays a major role. Can we have the overlap of boxes that show changes derived from different models/ ensemble. Higher the overlap, lower the changes (i.e., not statistically significant change). Such an approach is widely used in literature, as for example, <http://journals.ametsoc.org/doi/full/10.1175/JCLI3320.1>, where bootstrap was used to derive boxes. Here the boxes are automatically generated from models/ ensemble.

2. My other query is that if we compute the water availability based on P-E, that may provide a better information, and such P-E may be computed from atmospheric budget, using divergence. Such a method shows lower uncertainty (e.g., <https://www.nature.com/articles/srep29664>).

Otherwise, this is an excellent manuscript.

Subimal Ghosh
IIT Bombay.

Reviewer #3 (Remarks to the Author):

The study investigates the potential change in the risk of flood and drought exposure for a 1.5 and 2C global temperature increase. The authors highlight the vulnerability of the population living in the monsoon area and how they would be impacted under a warming scenario. The manuscript is well written and overall clear. I have no doubt that this study is important for both scientists and decision makers.

I have some major concern though to make it publishable in Nature Communication.

My main concern is about the use of RCP8.5 scenario and the way the authors select the 1.5 and 2C warming periods. The authors indicated they use a 10 years window for each period, and 20 years for the baseline. The main problem here is that a 10 years period may be very limited to remove the decadal oscillations. Thus the differences observed between 1.5 and 2C could be due to these oscillations (at least a part of it). I would suggest that the authors first quantify the decadal variability of the signal (with the baseline), and then compare it to the difference between 1.5 and 2C warming.

Moreover, the authors should check carefully the timing of the 1.5 and 2C periods in each model, to be sure they don't overlap (and if so, remove the corresponding models).

Ideally, it would have been better to use a lower level emission scenario (RCP2.6 for instance).

Also, as the paper is clearly about the differences between 1.5 and 2C warming in the monsoon regions, any discussion about other changes in temperatures or other regions should be removed

to make the communication more clear.

My other main concern is for the precipitation above 95th percentile. Does it include tropical cyclones too? It could be better to separate the contribution from TC and monsoon itself for region with high TC impact, or at least have some discussion on this point, as most of the CMIP5 models may not well represent the TC.

Finally, I think it could be worth it to improve the comparison between 1.5 and 2C periods by including also a comparison with a linear interpolation from 1.5C impact to 2C (basically, just multiply the results from 1.5C by a factor of $[2/1.5]$, and compare them with the actual modelled 2C warming). This would be a good way to discuss if the response in the extremes and vulnerability is purely linear to the temperatures or not.

Reply to Reviewer #1:

Thank you for the insightful comments and detailed instruction on how to improve the manuscript. The quality of the manuscript has been greatly improved based on your comments. In the following, the texts with *italic font* are your original comments, and the texts with normal font are our responses.

Reviewer #1 (Remarks to the Author):

Summary:

In the present manuscript, the authors quantify differences in precipitation characteristics focusing on the extreme tails across the world's monsoon regions for present-day climate, and hypothetical 1.5°C and 2°C worlds, including the impact of the half degree difference between the latter on precipitation characteristics. Subsequently, the authors go one step further towards quantifying "risk" by analyzing changes in precipitation characteristics jointly with population scenarios. I fully agree with the authors that understanding changes in precipitation characteristics, in particular with respect to the extreme tails of the distribution, is crucial; and a paper that carefully discusses and explains differences between a pre-industrial, and 1.5°C / 2°C worlds (and potentially higher-degree worlds), focusing on (extreme) precipitation induced risks, including a joint discussion/analysis with changes in exposure and/or vulnerability in the world's monsoon region could be an important contribution. Unfortunately, the present paper has little to offer in this regard due to

several shortcomings. This includes an unfocused discussion iterating some well-known results and subjective choices/statements, several conceptual issues related to the metrics used and how they are interpreted (i.e. drought vs. precipitation deficit metrics, and flood vs. high precipitation; but also the use of the concept of exposure and vulnerability as determinants of risk, see comments below), and finally issues of technical correctness of the paper related to deriving estimates based on a baseline climatology that could be prone to a potential overestimation of extremes outside the reference period, which requires some testing (please see detailed comments below). Due to these issues, I do not believe that the present manuscript can be considered for publication in the journal Nature Communications."

Response: Thank you for the insightful comments. The revision has been strongly guided by your comments which led to significant improvement of the manuscript. We have carefully investigated your comments. Here we summarize your two major concerns below along with an explanation of the major points of our corresponding revisions:

1) The selection of extreme precipitation metrics:

You argue that the threshold of 50% larger than the RX5day climatology might differ substantially across regions in terms of variability, and thus has limitations. We have performed further analysis and confirmed your hypothesis. Hence, in the revised manuscript, the widely used σ -exceedance approach, which measures changes with respect to existing variability (σ), is employed to define the ‘dangerous’ extreme events. In the Supplementary Information, we also compare the changes in exposure

if ‘dangerous’ extreme events are defined by 10- and 20-year return values in the 1950-2005 baseline, which is another widely used metric and is estimated from the generalized extreme value (GEV) distributions. The conclusion remains the same as that based on the σ -exceedance approach, increasing the fidelity of the results. For details, please see our responses to major comments #1(1) and #3 below.

2) Impact of potential artifacts induced by extreme estimators derived from a climatological baseline

In the revised manuscript, the σ -exceedance approach is used to measure the extreme precipitation changes, which requires normalization with respect to a baseline and, thus, may also be affected by the potential artifacts of extreme estimations for in- and out-of-base periods. We have tested whether the different base periods (1950-2005, 2006-2060, and 1950-2060) used for normalization alter our conclusions. The results show that the tests for all three baselines consistently reveal (1) increases in exposure to ‘dangerous’ extreme events with higher warming levels, which is associated with the rightward shift and broadening of probability density functions (PDFs) for extreme precipitation; and (2) robustly avoided impacts under the 1.5 °C warming scenario compared with 2 °C warming scenario. Thus, the key claims are not affected by the artifacts induced by normalization. For details, please see our responses to major comment #3 (i.e., technical correctness) below.

In addition to the above two major parts of the revision, we have made the following revisions based on your suggestions and comments:

1) We focus on changes in extreme precipitation and the discussion on droughts is deleted (response to major comment #2 (1));

2) Some conceptual issues such as flood vs. high precipitation (response to major comment #2 (1)), the concepts of risk/vulnerability/exposure (response to major comment #2 (2) and minor comment #6), the metric “change in hazard” × “population count” (response to major comment # 2 (2)) are clarified.

3) The physical reasons for changes in the mean and variability of extreme precipitation are discussed (response to major comments #1 (6) and #1 (2)).

4) We have deleted those unfocused discussions and the reiteration of previous results (response to major comments #1 (3), #1 (4), and #1 (5)).

Major comments:

1. Unfocused discussion in many places; lack of discussion of limitations and methodological description: While (again) the topic addressed by the authors is interesting, unfortunately the discussion of results is in many places unfocused, re-iterates several known results, and lacks explanations elsewhere (in particular as the authors present many results). Please find a few examples below:

(1) The paper relies on an extreme metric that the authors term 'unusual' or 'dangerous' wet extremes, which is defined as 50% greater intensities relative to the 1986-2005 baseline"(l. 20-21) in the Rx5day metric (the reader understands this only from the caption of Fig. 4; and I guess the authors mean 50% greater intensities relative to the RX5day mean of the 1986-2005 baseline", but this is not indicated).

However, it remains entirely elusive as to why the authors choose this metric; or why this metric might be important for impacts such as floods. While, clearly, 50% more than RX5day (mean) precipitation seems like a large amount, this might differ substantially across regions - i.e. there might be large gradients in the variability of RX5day across the GM regions. This means that while there might be some regions in which a 50% exceedance of the baseline is a 100-year event, in other regions a 50% exceedance might happen every other year and might thus not really be extreme (e.g. Hansen et al. 2012 discuss temperature extremes relative to their variability). Hence, I believe it would be crucial to discuss (a) why is a 50% exceedance relevant or meaningful?, and (b) what does 50% exceedance mean in terms of the variability of extremes? Please see other comments regarding the methodological approach further below.

Response: We investigate changes in the upper tail extremes, which are associated with the highest impacts, and thus termed as ‘dangerous’ events here. We fully agree that it is crucial to objectively define the threshold of ‘dangerous’. In the revision, we hope to employ the definition that has been widely accepted in the research community. We have reviewed the existing publications, there are three methods that are widely employed in studies on extreme climates and events:

(1) Signal-to-noise ratio (SNR; or termed as σ -exceedance). It measures the amplitude of changes in units of existing variability (σ ; e.g., Hawkins and Sutton, 2012; Hansen et al., 2012; Coumou and Robinson, 2013; Fischer et al., 2013; Lehner and Stocker, 2015; Frame et al., 2017). This has an advantage of understanding

climate change in the context of past experience, to which the human and ecological systems, as well as infrastructures have been adapted to. For instance, Frame et al. (2017) refers to climates with SNRs larger than 1, 2, 3 as ‘unusual’, ‘unfamiliar’, and ‘unknown’, respectively.

(2) Return value and return period. They measure the severity and rareness of an event (e.g., Kharin and Zwiers, 2005; Kharin et al., 2013).

(3) Percentage thresholds. It measures the amplitudes of events in percentages with respect to a reference climatology (e.g., Fischer et al., 2013).

In our original manuscript, we used the third method following Fischer et al. (2013). The “50% exceedance metric” means “50% greater intensities relative to the RX5day climatology of the 1986-2005 baseline”.

As you suggested, we examined what the 50% exceedance means in terms of the variability of extremes (Figure A1.1). To derive reliable extreme statistics, we calculate the variability as interannual standard deviation (σ) from detrended time series of extreme indices from 1950-2005. The results show that the meaning of “50% greater intensities” differs across regions in terms of variability. It ranges from 1 σ for the Australian monsoon region to 3 σ for part of the African and American monsoon regions. So as you argued, the uniform 50% exceedance of the baseline climatology corresponds to extreme events with diverse severity across regions, and is thus not appropriate for comparison across regions.

To overcome this limitation, in the revised manuscript we now use the first method, i.e., the σ -exceedance, as in many previous publications. We term extreme events

exceeding 1-4 σ as ‘dangerous’ events, representing different levels of severity and impacts. This threshold is more objective for comparison across regions. This method is introduced in detail in the Methods section of the revised manuscript (Please see L253-260). Please see the response to your comment #3 and Figs. 4-5 of the revised manuscript for the new results based on this method.

Figure A1.1 For RX5day, the ratio of 50% of the 1986-2005 climatology (threshold for the “50% exceedance metric”) and interannual standard deviation derived from detrended time series of 1950-2005. The multimodel medians are shown.

(2) The Abstract states that “Both the mean state and the variability of the extremes will intensify” (l. 15-16) as one of the major findings of the paper and shown also in Fig. 2d. However, this issues is left almost entirely out of the discussion in the main manuscript text. I believe a potential reader would like to understand if there are physical reasons for this kind of change, or whether this is just a statistical findings because mean and variance of extreme value distribution (unlike in a Gaussian

distribution, for instance) are not independent? Furthermore, it is not explained how the variability estimates are calculated.

Response: Thanks for your comments. Both changes in the mean state and variability of extreme precipitation are important for the frequency and associated changes in exposure to upper tail extremes, which is concerned in this study.

To depict the intensity of interannual variability, we calculate the interannual standard deviation (*std*) of the time series of extreme indices. Following Huntingford et al. (2013), a local detrending with an 11-year running mean is applied (i.e., removing the 11-year running mean from the original time series) to derive anomalies for each grid box, then the *std* is calculated over 20-year segments. The *std* over the GM region is then calculated with area weighting.

The results show that the interannual variability of extreme precipitation increases with warming levels in multimodel simulations (blue curve in Fig. 2c in the revised manuscript). This finding is independent of the detrending methods. If the time series is linearly detrended first, and then applied with an 11-year high-pass filter to obtain the component of interannual variability, based on which the *std* is calculated, similar results are yielded (see Supplementary Fig. S8c).

To understand the underlying physics, we first examine the role of thermodynamics by assuming an idealized enhancement due to the Clausius-Clapeyron relationship, following the approach of Allan and Soden (2008). Here the extreme precipitation events at the 1.5°C, 2°C, 3°C, and 4°C are theoretically calculated by an intensification of 7%/K relative to the 0°C warming reference (1866-1885, the central

20-year period in the 1861-1890 pre-industrial period). This results in a theoretical increase in the variability by 7%/K (red curve in Fig. 2c in the revised manuscript). Compared to this thermodynamical hypothesis, a larger increase in variability in extreme precipitation is found in model simulations, by 9.05%/K with the 25-75th range of 7.90-11.14%/K (cf. blue and red curves in Fig. 2c in the revised manuscript). In this hypothesis, a uniform increase of mean moisture is assumed, without considering changes in the variability of moisture. Actually, associated with the saturated atmospheric moisture increasing by 7%/K, a simultaneous increase in the variability at the same rate is obtained theoretically. The simulated increasing variability of moisture is generally consistent with the thermodynamic arguments (Fig. 2d in the revised manuscript). Hence, the increased variability of moisture further enhances the associated variability in extreme precipitation.

We explain the calculation of variability in the Methods section (Please see L226-236) and discuss the changes in the variability of extremes in Fig. 2 and L93-109 of the revised manuscript.

(3) l. 120-126, the authors describe an extreme flooding event that occurred in India in 2013, which had large negative impacts. However, first, this discussion falls out of the blue as the event is arbitrarily chosen, and does not in any way that it obvious to the reader connect to the more large-scale analysis conducted in the remainder of the paper. Second, and this issue follows up on the Rx5day variability estimates (see my comment above), the authors state: “Thus, we emphasize that future changes in the

variability of extremes are as important as the changes in the mean state, as an increase in either can induce more unusual and damaging extreme events.” (l. 126-129) While this statement seems broadly plausible (but is not based on an analysis), it remains entirely dubious to me why the authors conclude this from a single case study, or in which other way it might be linked to this discussion.

Response: Thanks for your comment. We agree that it is not rigorous to demonstrate the importance of variability based on a single case and thus the statement has been deleted in the revision. Instead, the importance of changes in the mean and variability in inducing upper tail extremes is now highlighted by referring to some famous publications (e.g., Katz and Brown, 1992; Zhang and Zwiers, 2013). Please see L136-137 of the revised manuscript.

(4) l. 86-95 The authors devote quite a long discussion to the relationship between $RX5day$ and $R90ptot$, $R95ptot$, and $R99ptot$ - I wonder what is the purpose of this discussion, triggered with "Here we show" in l.86; but not at all mentioned in the Abstract or the objectives of the paper? How does this discussion help to addressing the objectives of the paper? (It is not surprising that these metrics are correlated. Moreover, the correlations are clearly non-linear (Fig. 2c) - hence, fitting a linear regression lines is clearly inappropriate...)

Response: Thanks for the suggestion. We agree that it is not surprising to see that these metrics are correlated. This part is now deleted in the revision and we focus on the changes of $RX5day$.

(5) l. 91-93 It has been shown many times in the literature that GCM simulations under climate change tend to produce increases in precipitation extremes, but decreases in the lower part of the distribution. Of course, it is ok to state this again (l. 91-93), or to present this again (Fig. 2a), but then appropriate original literature should be cited (for instance see Allen and Ingram 2002).

Response: Accepted. Original literatures are cited in the revised manuscript (e.g., Trenberth et al., 2003; Allan and Soden, 2008; Please see L78 and L239 in the revised manuscript).

Since literatures have noted that GCM simulations under climate change tend to produce increases in precipitation extremes, but decreases in the lower part of the distribution, the response of precipitation structure to global warming (Fig. 2a-b in the original manuscript) is now deleted in the revised manuscript.

(6) l. 81: The authors broadly refer to the Clausius-Clapeyron relationship and "dynamic" changes as causes for these observed changes in precipitation extremes - which is certainly correct but seems overly simplistic and rather a standard explanation. Since the authors are focusing on the GM regions, I believe a reader would be interested to understand if there are mechanisms that could change the dynamics in a certain way in the GM regions? I do not think a detailed analysis would be needed, but a few sentences more about the underlying reasons of dynamical changes, and what is specific to global GM regions, would be clearly necessary.

Response: Thanks for your suggestion. In the revised manuscript, we add the following discussions on possible dynamical processes affecting changes in extreme precipitation over the monsoon regions (Please see L78-86):

The weaker response of the RX5day in simulations compared to the thermodynamic arguments implies a potential offset from the dynamic changes (blue vs. red curves in Fig. 2a). Changes in dynamic circulation can substantially affect extreme precipitation¹⁹⁻²¹. The underlying mechanisms include the weakening of the large-scale monsoon circulation^{11, 22-25}, which results from the stabilization of the tropical atmosphere associated with global warming²⁶⁻²⁷, the modulation of regional monsoon circulation by land-sea thermal contrast changes²⁸⁻²⁹, the gradient of sea surface temperature warming patterns³⁰, and changes in synoptic scale circulations, such as monsoon depressions^{21, 29, 31} and tropical cyclones³².

(7) The issues discussed above lead to a paper that does not appear to be very focused towards achieving certain objectives, and as a consequence does not seem to present or explain novel findings beyond that extremes in the GM regions are intensifying in global model simulations.

Response: Thanks for the comments. In the revised manuscript, the reiteration of well-known results, the changes in precipitation structure and CDD are deleted (Please see response to your comment #2 (1) in detail). Guided by your and the other two reviewers' suggestions, we focus on the extreme precipitation changes associated with the half a degree less or additional warming. We highlight our major findings in

the revision, including (1) the nonlinear increases in exposure to ‘dangerous’ extreme precipitation events with higher warming levels, associated with both increases in the mean and variability of extreme precipitation; (2) robustly avoided impacts at the 1.5°C warmer world compared with 2°C; (3) regional hotspots impacted prominently by the 0.5°C less or additional warming.

Please also see our summary of the major revisions given in the beginning of the response letter. Thank you for the constructive criticism that has led to a very focused manuscript towards quantifying and understanding the changes in exposure to extreme precipitation associated with the half a degree less or additional warming over the populous global land monsoon region.

2. Conceptual issues and methodological limitations need to be discussed:

(1) Drought vs. CDDs and Extreme Precipitation vs. “Flood”

The title of the paper is “Extra 0.5°C warming increases exposure to floods and droughts in global land monsoon regions”. The authors acknowledge briefly in l. 66-67 that their analysis only deals with the precipitation-related components of flood and drought risks; however, the broader implications of this limitation for the analysis would need to be discussed, in particular if the paper is intended for a broad readership journal. For example, even disregarding the impact on people (vulnerability and exposure; see comment below), the physical origins of droughts are by far not only determined by CDD. See for instance Sheffield et al. 2012; which explains in large detail that there are various feedback mechanisms at play (briefly

mentioned by the authors, l. 57-61), but not all of these need to be negative in a warming climate (Seneviratne et al. 2012). Hence, based on the analysis conducted, it is inappropriate to claim that droughts are increasing across global monsoon regions. Partly, the problem might stem from the fact that the authors aim to cover droughts and floods in a single paper, which leaves relatively little room to give an indication of the drivers, let alone broader implications of changes in either phenomenon (in particular such a complex phenomenon such as drought).

Response: Thanks. As you suggested, we should not hope to cover both floods and droughts in a single paper with limited space, especially given the complex processes of droughts. In our revision, the analysis of droughts is deleted and we focus on the changes in extreme precipitation. The indication of the drivers and implications of increases in exposure are discussed.

(2) Eventually, the authors address “climate risk” related to “droughts” and “floods” (l. 35-38). Risk is typically determined as a function of hazard, vulnerability, and exposure (IPCC, 2012); so I am wondering why the authors only mention hazard and vulnerability in their discussion in l. 35-38; but then address hazard and exposure in their analysis (Fig. 3+4). On another note, while I agree that understanding the dynamics in people exposed to any kind of climate extreme is important in the future (which the authors address towards the end of the paper; e.g. in Fig. 4), it seems unclear to me why the metric “Change in hazard” x “Population count” as addressed in Fig. 3 would be particularly meaningful (-> would big changes in extremes in a

sparsely populated place be not important to address?) Please explain/discuss (a) why this metric is relevant, (b) its limitations, and (c) why the relationship between hazard and exposure would be simply multiplicative (and not, for instance highly non-linear towards big population changes and/or climate changes).

Response:

According to IPCC SREX (2012), risk is determined by hazard, vulnerability and exposure of human society and natural ecosystems. In addition, vulnerability itself is a function of exposure, sensitivity, and adaptive capacity (IPCC SREX 2012). In this work, we quantify the exposure component of vulnerability to extreme precipitation, which is fundamental for the understanding of future vulnerability and the development of mitigation and adaptation strategies. We have clarified the terms of ‘risk’, ‘vulnerability’, and ‘exposure’ in the revised manuscript (Please see L58-63).

Below are the responses to your questions on the metric “Change in hazard” × “Population count”:

(a) Why is this metric relevant?

The metric “Change in hazard” × “Population count” has been widely used in the climate research community, as an efficient way to understand how exposure emerges as a result of both climate change and population distribution (e.g., Sedláček and Knutti, 2014; Jones et al., 2015; Frame et al., 2017). This metric is used here since the global monsoon regions are also densely populated regions. In the revision, we first examine the changes in extreme precipitation at different warming levels (Fig. 3a-c in the revised manuscript). Based on the changes in extreme precipitation, we further

quantify their impacts on population groups using the metric “Change in hazard” × “Population count”. This metric highlights the impacts of climate change on the human groups, in particular over the densely populated monsoon regions. The interpretation of this metric should be based on the change patterns of hazards, i.e., extreme precipitation in this study.

In addition, the spatial patterns of this metric reveal hotspots that suffer great impacts, from either large increases in hazards or dense populations, or both. Change patterns of this metric between different warming levels indicate where the low warming target has the largest impact (Fig. 3d-f in the revised manuscript).

(b) Its limitations

Large values in this metric require that neither “Change in hazard” nor “population count” can be too small. Thus, it does not reveal big changes in hazards in a sparsely populated place or dense population experiencing very weak changes in hazards, as you noted. Thus in the revised manuscript, we emphasize that the interpretation of this metric should be based on the changes in hazards (Please see L251-252 in the revised manuscript).

(c) Why the relationship between hazard and exposure would be simply multiplicative?

The metric “Change in hazard” × “population count” is identical in spatial pattern to “population weighted changes in hazards”, as we interpreted in the revised manuscript (Please see L113-117). The latter can be expressed as “Change in hazard” × “population count” / “total population”.

The above concerns are highlighted and clarified in the revised manuscript (Please see L110-122 and L247-252).

3. Technical Correctness

Extreme indicators derived from percentile estimators of a baseline climatological period are known to be sensitive to an overestimation of extremes in the period outside of the baseline period (see e.g. Zhang et al., 2005), because a 20-year or 30-year baseline period is often too short to derive reliable estimates of extreme statistics. This is also the case for other metrics that are estimated from a climatological baseline (see e.g. Sippel et al. 2017) if aggregated across space, such as wet extremes with 50% greater intensities relative to the 1986-2005 baseline." Percentile-based indicators are used extensively in the paper (Fig. 2a-c, for instance) as well as the 50% metric"(Fig. 4 and highlight statements in the Abstract), but there is no indication whether the authors have corrected for this potential artefact, or whether the authors have tested if this plays a role here. Although I am not sure whether these potential artefacts are relevant here (they might be small; but can be severe sometimes), I believe it would be crucial to test for these issues (in particular because the 50% metric seems quite arbitrary as discussed above).

Response: Thanks for your comments and suggestions. The technical correctness has greatly enhanced the robustness of our results. We address this issue in the following two aspects.

(1) Percentile-based indicators

As you noted, the conventional method of calculating percentiles based on a fixed base period would induce an artificial jump of exceedance frequency between the in-base and out-of-base periods, because of sampling error in calculating thresholds for the base period. Thus, it may not be suitable for trend analysis (Zhang et al., 2005). In the original manuscript (Fig. 2a-c of the original manuscript), we examined the responses of precipitation structure and high-percentile precipitation to temperature increase based on percentile thresholds estimated from the 1986-2005 base period. We then calculate the changes of percentile exceedance in 2006-2100 in projections, which lies totally out of the base. Thus, the results are not affected by the artificial jump between the in-base and out-of-base periods. Indeed, as you noted, a 20- or 30-year baseline period is often too short to derive reliable estimates of extreme statistics. Thus it would be better to estimate the percentile thresholds using longer base periods, e.g., 1950-2005, or hundreds of years in piControl simulations.

Based on your comments #1(4), #1(5) and #1(7), to be more focused on our objective, viz. extreme precipitation, we delete the discussions on precipitation structure in the revised manuscript.

(2) From the “50% exceedance” metric to “ σ -exceedance” metric

As noted in the response to your comment #1(1), we change the “50% exceedance” metric to σ -exceedance threshold in the revision, which is more objective for comparison across regions. To derive reliable extreme statistics, in the revision, a 56-year long base period 1950-2005 is used to estimate the mean and standard deviation (σ) for normalization. Firstly, it is a period with relatively good global

coverage of observations to constrain the models. Secondly, it is a climate that the human systems are familiar with and adapted to.

Considering that conventional normalization relative to the local mean and variability of a reference period changes the distributions differently for the in-base and out-of-base periods, and thus affect the frequency of extremes (Sippel et al., 2015, 2017), we tested whether this affects our conclusions. We compare the results using alternative base periods of 2006-2060 (middle column in Figure A1.2) and 1950-2060 (right column in Figure A1.2) for normalization. We ended the base periods in 2060 since the latest timing of a 2°C warming is 2053 for MRI-CGCM3 (see Supplementary Table S1). Specifically, we address the following three issues: 1) How does the distribution of extreme indices change with global warming? 2) How does exposure to ‘dangerous’ events change with warming? 3) How does the selection of baseline affect the results regarding the avoided impacts at the 1.5°C warming compared to 2°C in terms of exposure to ‘dangerous’ extreme events?

1) How does the distribution of extreme indices change with global warming?

The PDFs of normalized RX5day shift rightward and stretch especially at the upper tails, as the climate warms (Figure A1.2, upper row). When a later period 2006-2060 is included in the reference (i.e., the 2006-2060 and 1950-2060 cases), the changes in PDFs are less prominent compared with the 1950-2005 case. This is because the mean and standard deviation of extremes increase with warming, when applying the normalization ($z = \frac{X-\mu}{\sigma}$), large anomalies are actually “minused” and “divided out” if a later period is included in the baseline. Such characteristics have been documented

in Hansen et al. (2012) and Trenberth et al. (2013). It is thus demonstrated that including climate change effects in the baseline ‘alters the ranges of observed variability against which the longer-term variations that characterize changes are scaled’, and thus ‘greatly reduces any prospects of identifying a climate change signal’ (Trenberth et al., 2013).

2) How does exposure to ‘dangerous’ events change with warming?

‘Dangerous’ events, defined as events exceeding $1-4\sigma$, is associated with the upper tails of the PDFs. Both the rightward shift and broadening of the PDFs lead to more area exposed to these extreme events as the climate warms (Figure A1.2, middle row). It is qualitatively consistent among the three baselines (viz. 1950-2005, 2006-2060, and 1950-2060). Quantitatively, the area of exposure is the largest when 1950-2005 is used as the baseline, and is the smallest for the 2006-2060 case. This is probably because a 1σ exceedance event is much stronger in the 2006-2060 case than the 1950-2005 case, and thus the quantitative exposures among the three cases are not directly comparable.

3) How does the selection of baseline affect the results regarding the avoided impacts at the 1.5°C warming compared to 2°C in terms of exposure to ‘dangerous’ extreme events?

The avoided impacts by the 0.5°C less warming are quantified as the difference of impacts at the 2°C and 1.5°C warming levels (see Methods in the revised manuscript; Figure A1.2, bottom row). For the baseline of 1950-2005, both the 1.5°C and 2°C periods lie out of the base; while for the baselines of 2006-2060 and 1950-2060, both

the 1.5°C and 2°C periods lie in the base. Hence, no artificial jumps exist between the 1.5°C and 2°C conditions. The major conclusion, i.e., the reduced exposure to ‘dangerous’ events associated with the half a degree less warming, are robust for the three baselines, although we should not quantitatively compare their magnitudes .

In summary, our key claims are robust against the alternative selections of baseline. In the manuscript, we use the 1950-2005 period as the baseline for normalization (Please see L253-257 in Methods in the revised manuscript).

In addition, we have also examined the avoided impacts by the 0.5°C less warming based on the extreme value theory. We compared the changes in exposure if the ‘dangerous’ extreme events are defined by the 10- and 20-year return values in the 1950-2005 baseline, estimated from the generalized extreme value (GEV) distributions by the method of maximum likelihood. The results show that the land area exposed to the baseline 10- and 20-year return values consistently increases with warming. A ~11% reduction of exposure (with an interquartile range of 3-20%) relative to the present-day level for both return values is expected for the 1.5°C warming compared with 2°C (Supplementary Fig. S4). The results based on return values enhance the robustness of our conclusions, and are noted in L173-175 of the revised manuscript and Supplementary Texts and Fig. S4.

Figure A1.2 Comparison between the three base periods used for normalization, i.e., 1950-2005 (left column), 2006-2060 (middle column), and 1950-2060 (right column). Top, PDFs of land fraction of RX5day anomalies in units of σ aggregated over the GM region at the pre-industrial (pI, 1861-1890), present-day (pd, 1986-2005), 1.5°C, 2°C, 3°C, and 4°C warming levels. Middle, area exposed to RX5day events exceeding 1-4 σ at different warming levels. Bottom, reduced area of exposure to 1-4 σ exceedance events at the 1.5°C warming compared to 2°C (% relative to the present-day level).

Minor comments:

1. l. 44: “nearly two thirds” -> why not state directly that it is 61.8%?

Response: Accepted. Changed as “~62% of the world’s population” in the revised manuscript (Please see L43 in the revised manuscript).

2. l. 72: How was the model selection done? Based on an ensemble of opportunity?

Response: Thank you for your questions. The model selection is done based on (1) daily precipitation data availability; (2) a 2°C warming is arrived before the year 2100 under both RCP4.5 and RCP8.5 (the results based on RCP4.5 is compared at the end); (3) the difference in the timings of a 1.5 and 2°C warming is no less than 9 years, so as to avoid overlaps between the two conditions since 9-year time windows are used to represent respective levels (as suggested by Reviewer #3).

We have clarified the model selection in the Methods section in the revised manuscript (Please see L187-191).

3. l. 104: Are there some models in the multi-model ensemble that show lower or higher sensitivities? Would it be possible to derive across-model confidence intervals?

Response: Thanks for your suggestion. The inter-model spread in changes in extreme precipitation and associated exposure is shown in Figs. 1, 2, 4, 5 of the revised manuscript and stated throughout the text (e.g., L22, L72-73, L141-142, etc.).

4. l. 121-122: you state “history has told that extreme events that deviate largely from

the climatology can be the most destructive”. In the introduction, and scattered throughout the paper, you state that exposure and vulnerability are also crucial contributors to the impacts of climate extremes. Why is this not so here?

Response: Climate risk is typically determined by hazard, vulnerability and exposure of human society and natural ecosystems (IPCC SREX 2012). The impacts induced by hazards are determined by both the climate and societal factors. In this study, we only focus on the impacts induced by climate changes. The statement “*history has told that extreme events that deviate largely from the climatology can be the most destructive*” hoped to highlight the impact of extreme climate regardless of changes in societal factors like vulnerability.

We have clarified this in L123-127 of the revised manuscript.

5. l. 134-135: “50% greater intensities relative to the baseline climatology”: Please clarify which metric (I am assuming Rx5day) and whether you mean the mean baseline climatology.

Response: In the original manuscript, we meant “50% greater intensities relative to the baseline (1986-2005) climatology of RX5day”. In the revised manuscript, we have changed this metric to the σ -exceedance approach as a more objective one (Please see responses to your comments #1(1) and #3 in detail).

6. L136-138: According to standard literature in the field (e.g. IPCC 2012), vulnerability and exposure are distinct features: Please clarify the terminology.

Response: Accepted. Please see L58-60 of the revised manuscript for clarification.

References:

Allan, R. P., & Soden, B. J. Atmospheric warming and the amplification of precipitation extremes. *Science*, **321**, 1481 (2008).

Coumou, D., & Robinson, A. Historic and future increase in the global land area affected by monthly heat extremes. *Environ. Res. Lett.* **8**, 34018-34016 (2013).

Fischer, E. M., Beyerle, U., & Knutti, R. Robust spatially aggregated projections of climate extremes. *Nat. Clim. Change* **3**, 1033-1038 (2013).

Frame, D., Joshi, M., Hawkins, E., Harrington, L. J., & Roiste, M. D. Population-based emergence of unfamiliar climates. *Nat. Clim. Change* **7**, 407-411 (2017).

Hansen, J., Sato, M., & Ruedy, R. Perception of climate change. *Proc. Natl Acad. Sci.* **109**, E2415-E2423 (2012).

Hawkins, E., & Sutton, R. Time of emergence of climate signals. *Geophys. Res. Lett.* **39** (2012).

Huntingford, C., Jones, P. D., Livina, V. N., Lenton, T. M., & Cox, P. M. No increase in global temperature variability despite changing regional patterns. *Nature*, **500**, 327 (2013).

IPCC: *Managing the Risks of Extreme Events and Disasters to Advance Climate Change Adaptation*. A Special Report of Working Groups I and II of the Intergovernmental Panel on Climate Change [Field, C.B., V. Barros, T.F. Stocker, D. Qin, D.J. Dokken, K.L. Ebi, M.D. Mastrandrea, K.J. Mach, G.-K. Plattner, S.K. Allen, M. Tignor, and P.M. Midgley (eds.)]. Cambridge University Press, Cambridge, UK,

and New York, NY, USA (2012).

Jones, B., O'Neill, B. C., McDaniel, L., McGinnis, S., Mearns, L. O., & Tebaldi, C. Future population exposure to US heat extremes. *Nat. Clim. Change* **5**, 652-655 (2015).

Katz, R. W., & Brown, B. G. Extreme events in a changing climate: variability is more important than averages. *Climatic Change*, **21**, 289-302 (1992).

Kharin, V. V., & Zwiers, F. W. Estimating extremes in transient climate change simulations. *J. Clim.* **18**, 1156-1173 (2005).

Kharin, V. V., Zwiers, F. W., Zhang, X. et al. Changes in temperature and precipitation extremes in the CMIP5 ensemble. *Climatic Change* 119-345. (2013)

Lehner, F., & Stocker, T. F. From local perception to global perspective. *Nat. Clim. Change* **5**, 731-734 (2015).

Sedláček, J., & Knutti, R. Half of the world's population experience robust changes in the water cycle for a 2 °C warmer world. *Environ. Res. Lett.* **9**, 044008 (2014).

Sippel, S., et al. Have precipitation extremes and annual totals been increasing in the world's dry regions over the last 60 years? *Hydrol. Earth Syst. Sci.* **21**, 441-458 (2017).

Sippel, S., et al. Quantifying changes in climate variability and extremes: Pitfalls and their overcoming. *Geophys. Res. Lett.* **42**, 9990-9998 (2015).

Reviewer #2 (Remarks to the Author):

This is a very interesting and timely work and may serve as a very important document for policy making. In my opinion, it is worth publishing such an article.

However, I have couple of points, which the authors may address:

Response: Thank you for the insightful suggestions and detailed instruction on how to improve the manuscript. In the following, *the texts with italic font are the reviewer's original comments*, and the texts with normal font are the authors' responses.

1. There is significant uncertainty across models, and hence we get a box plot of changes derived from the models. Under such an uncertain scenario, statistical significance plays a major role. Can we have the overlap of boxes that show changes derived from different models/ ensemble. Higher the overlap, lower the changes (i.e., not statistically significant change). Such an approach is widely used in literature, as for example, <http://journals.ametsoc.org/doi/full/10.1175/JCLI3320.1>, where bootstrap was used to derive boxes. Here the boxes are automatically generated from models/ ensemble.

Response: Thank you for your suggestion. The uncertainty in changes in extreme precipitation has been recognized both in theory and model projections (Fischer et al., 2013; O’Gorman 2015; Pfahl et al., 2017). Basically, there are three categories of methods widely used to denote the statistical significance or robustness of changes in

multi-model/ensemble simulations:

(1) Agreement in the sign of change across models. This method has been extensively used in the IPCC fourth Assessment Report (e.g., IPCC WG1 AR4, chapter 10 and 11), fifth Assessment Report (e.g., IPCC WG1 AR5, chapter 14), IPCC SREX Report (e.g., chapter 3), and recent literatures (e.g., Li et al., 2015; Chleussner et al., 2016; Pfahl et al., 2017; Lau and Kim, 2017).

(2) Multimodel ensemble mean (or median) changes vs. climate internal variability (e.g., Tebaldi et al., 2011; Power et al., 2012; Sedláček and Knutti, 2014). Usually climate internal variability is estimated from interannual variations in the base period or pre-industrial control runs (Collins et al., 2013).

(3) Multimodel ensemble mean (or median) changes vs. model spread. This method examines the robustness of changes against model uncertainty. Here Model spread is usually measured by standard deviations across models or ensembles (e.g., Pendergrass et al., 2015; Sanderson et al., 2017).

In our analysis, since we hope to contribute this investigation to the IPCC Special Report on the 1.5 °C warming, to facilitate the comparison with other results presented in the report, robustness of changes between the 1.5 °C and 2 °C warming levels here is indicated based on model agreement in the sign of change, viz. method (1), as is widely used in IPCC reports. The method of Kharin and Zwiers (2005) is usually used to estimate significance of forced signals versus the climate internal variability in the ensemble simulations of individual model. In their simulation, since they only have 3

ensemble members of the Canadian model, the 10%-90% confidence intervals are obtained via “bootstrapping” by random resampling, and thus, the confidence intervals measure the climate internal variability, rather than the model spread. We hope to use it in our future studies focusing on comparison between forced responses and internal variability, based on individual model with large ensemble simulations.

2. My other query is that if we compute the water availability based on P-E, that may provide a better information, and such P-E may be computed from atmospheric budget, using divergence. Such a method shows lower uncertainty (e.g., <https://www.nature.com/articles/srep29664>).

Response: Thank you for your suggestions. We investigated the changes in P-E based on your suggestion (Figure A2.1). The results are interesting, and several related issues warrant further investigation, e.g., the hotspots impacted by water availability changes, and uncertainty in projections.

Due to the limited room of the current manuscript, the revised manuscript focuses on extreme precipitation. We hope to report the results on water availability in a separate study with further in-depth research. In L180-182 of the revised manuscript, we highlight this important topic and call for further study by citing the recommended literature.

Figure A2.1 Multimodel median changes in P-E (a-c) and atmospheric moisture convergence (d-f), from the present-day (1986-2005) to 1.5 °C warming (a, d), the present-day to 2 °C warming (b, e), and 1.5 °C to 2 °C warming (c, f). Stippling denotes where at least 2/3 of the models agree in the sign of change.

References:

Christensen, J. H. et al. 2007: Regional climate projections. In: *Climate Change 2007: The Physical Science Basis. Contribution of Working Group I to the Fourth Assessment Report of the Intergovernmental Panel on Climate Change*. Cambridge University Press, Cambridge, United Kingdom and New York, NY, USA, pp. 847–940.

Christensen, J.H. et al. 2013: Climate Phenomena and their Relevance for Future Regional Climate Change. In: *Climate Change 2013: The Physical Science Basis. Contribution of Working Group I to the Fifth Assessment Report of the Intergovernmental Panel on Climate Change*. Cambridge University Press, Cambridge, United Kingdom and New York, NY, USA.

Collins, M., et al. 2013: Long-term Climate Change: Projections, Commitments and Irreversibility. In: *Climate Change 2013: The Physical Science Basis. Contribution of Working Group I to the Fifth Assessment Report of the Intergovernmental Panel on Climate Change*. Cambridge University Press, Cambridge, United Kingdom and New York, NY, USA.

Fischer, E. M., Beyerle, U. & Knutti, R. Robust spatially aggregated projections of climate extremes. *Nat. Clim. Change*. **3**, 1033-1038 (2013).

Kharin, V. V., and Zwiers, F. W. (2005). Estimating extremes in transient climate change simulations. *Journal of Climate*, *18*(8), 1156-1173.

Lau, K. M., & Kim, K. M. (2017). Competing influences of greenhouse warming and aerosols on Asian summer monsoon circulation and rainfall. *Asia-Pacific Journal of Atmospheric Sciences*, 53(2), 181-194.

Li, X., Ting, M., Li, C., & Henderson, N. (2015). Mechanisms of asian summer monsoon changes in response to anthropogenic forcing in CMIP5 models. *Journal of Climate*, 28(10).

Meehl, G.A. et al. 2007: Global Climate Projections. In: *Climate Change 2007: The Physical Science Basis. Contribution of Working Group I to the Fourth Assessment Report of the Intergovernmental Panel on Climate Change*. Cambridge University Press, Cambridge, United Kingdom and New York, NY, USA.

O’Gorman, P. A. (2015). Precipitation extremes under climate change. *Current climate change reports*, 1(2), 49-59.

Pendergrass, A. G., Lehner, F., Sanderson, B. M., & Xu, Y. Does extreme precipitation intensity depend on the emissions scenario? *Geophys. Res. Lett.* **42**, 8767-8774 (2015).

Pfahl, S., O’Gorman, P. A., & Fischer, E. M. Understanding the regional pattern of projected future changes in extreme precipitation. *Nat. Clim. Change* **7** (2017).

Power, S. B., Delage, F., Colman, R., & Moise, A. (2012). Consensus on twenty-first-century rainfall projections in climate models more widespread than previously thought. *Journal of Climate*, 25(11), 3792-3809.

Sanderson, B. M. et al. (2017). Community climate simulations to assess avoided impacts in 1.5 and 2 °C futures. *Earth System Dynamics*, 8(3), 827.

Schleussner et al. Differential climate impacts for policy-relevant limits to global warming: the case of 1.5 °C and 2 °C. *Earth Syst. Dyn.* **7**, 327-351 (2016).

Sedláček, J., & Knutti, R. Half of the world's population experience robust changes in the water cycle for a 2 °C warmer world. *Environ. Res. Lett.* **9**, 044008 (2014).

Seneviratne, S.I. et al. Changes in climate extremes and their impacts on the natural physical environment. In: *Managing the Risks of Extreme Events and Disasters to Advance Climate Change Adaptation*. A Special Report of Working Groups I and II of the Intergovernmental Panel on Climate Change (IPCC). Cambridge University Press, Cambridge, UK, and New York, NY, USA, pp. 109-230 (2012).

Tebaldi, C., Arblaster, J. M., & Knutti, R. (2011). Mapping model agreement on future climate projections. *Geophysical Research Letters*, 38(23).

Reviewer #3 (Remarks to the Author):

The study investigates the potential change in the risk of flood and drought exposure for a 1.5 and 2°C global temperature increase. The authors highlight the vulnerability of the population living in the monsoon area and how they would be impacted under a warming scenario. The manuscript is well written and overall clear. I have no doubt that this study is important for both scientists and decision makers.

I have some major concern though to make it publishable in Nature Communication.

Response: Thank you for the insightful comments and detailed instruction on how to improve the manuscript. The quality and clarity of the manuscript have been greatly improved based on your comments. In the following, *the texts with italic font are the reviewer's original comments*, and the texts with normal font are the authors' responses.

1. My main concern is about the use of RCP8.5 scenario and the way the authors select the 1.5 and 2°C warming periods. The authors indicated they use a 10 years window for each period, and 20 years for the baseline. The main problem here is that a 10 years period may be very limited to remove the decadal oscillations. Thus the differences observed between 1.5 and 2°C could be due to these oscillations (at least a part of it). I would suggest that the authors first quantify the decadal variability of the signal (with the baseline), and then compare it to the difference between 1.5 and 2°C warming.

Response: Thank you for your comments. In this study, the differences between the 1.5

and 2°C warming are shown in multimodel ensemble medians. The ensemble median approach is supposed to minimize the influence of climate internal variability, including decadal oscillations.

To address the role of decadal oscillations in the long-term evolution of extreme precipitation, we apply an Empirical Orthogonal Function (EOF) analysis to the 9-year running average extreme indices over the GM region for multi-model ensemble median and individual models, respectively (Supplementary Figure S2). For both multimodel median and individual models, the leading modes feature a spatially consistent increase, reflecting the effect of global warming, and explaining up to 90% and around 40% of the total variance, respectively. Thus the effect of global warming well dominates over decadal oscillations in the evolution of 9-year smoothed extreme precipitation.

To further investigate whether decadal variability affects the signals derived (i.e., the differences between the 1.5 and 2°C warming levels), we compare the results derived from time series with and without decadal oscillations (Figure A3.1). To remove the decadal oscillations, a 10-70-year band pass filter is applied and then removed from the original time series. Thus only the interannual variability and long-term trend remains. We use the lowest frequency of 70-year for filtering since periods of 10-70 years are dominant for AMO (Atlantic Multidecadal Oscillation) in CMIP5 models (Han et al., 2016).

The changes in extremes between different warming levels (i.e., the signals) from the original time series are shown in Figure A3.1a. The differences in signals between the

original and filtered cases can be regarded (at least partly) as the effect of the remaining decadal variability from 9-year averages (Figure A3.1b). The results show that the influence of the remaining decadal oscillations on ensemble median changes is small, typically lower than 1% of the 1986-2005 present-day level in most cases and most monsoon regions. Some larger differences (still lower than 3% of the present) are seen in the North American and Australian monsoon regions, partly due to their small land coverages and small signal-to-noise ratio (SNR, ensemble median changes / inter-model standard deviations; Figure A3.1c). The SNRs in the original cases are overall slightly lower than in the filtered cases, partly due to additional noises induced by the remaining decadal oscillations.

In summary, while decadal variability may potentially lower the SNR, the multimodel ensemble approach has largely eliminated the effect of the remaining decadal variability in terms of signal. We have clarified the impact of decadal variability in the Methods section in the revised manuscript (Please see L210-215 and Supplementary Fig. S7).

Figure A3.1 (a) Multimodel median changes (% with respect to the 1986-2005 present-day level) in RX5day for the GM and individual monsoon regions, from the present day to 1.5°C (blue), present day to 2°C (pink), and 1.5°C to 2°C (green), derived from the original time series. Bars denote one standard deviation across models. (b) Differences (% with respect to the 1986-2005 present-day level) in ensemble median changes derived from the original and filtered time series. (c) Signal (multimodel median changes) to noise (inter-model standard deviation) ratio (SNR). Bars represent the filtered cases while diamonds represent the original ones.

2. Moreover, the authors should check carefully the timing of the 1.5 and 2°C periods in each model, to be sure they don't overlap (and if so, remove the corresponding models).

Response: We checked the difference of timings of the two levels in Table A3.1. Since a 9-year period centered on the specific year is referred to as the 1.5 or 2°C condition, non-overlapping requires the difference between the two timings to be no less than 9 years ($\Delta T \geq 9\text{yr}$). Thus for the RCP8.5 (RCP4.5) emission scenario, FGOALS-s2 and CSIRO-Mk3-6-0 (FGOALS-s2) should be excluded. In the revised manuscript, we use the remaining 21 models for analysis.

Selection of models is introduced in the Methods section of the revised manuscript (Please see L188-191).

Table A3.1 The timings (in year) of 1.5 and 2°C warming and their differences under RCP4.5 and RCP8.5, respectively.

Model	RCP4.5			RCP8.5		
	1.5°C	2°C	ΔT	1.5°C	2°C	ΔT
ACCESS1-0	2030	2052	22	2025	2042	17
ACCESS1-3	2038	2055	17	2031	2042	11
bcc-csm1-1	2022	2043	21	2021	2036	15
bcc-csm1-1-m	2015	2035	20	2012	2028	16
CanESM2	2018	2032	14	2013	2027	14
CCSM4	2017	2040	23	2015	2029	14
CESM1-BGC	2021	2044	23	2017	2034	17

CMCC-CM	2035	2052	17	2030	2040	10
CMCC-CMS	2033	2053	20	2029	2042	13
CNRM-CM5	2035	2056	21	2029	2043	14
CSIRO-Mk3-6-0	2035	2049	14	2035	2043	8
FGOALS-s2	2010	2012	2	2010	2012	2
GFDL-CM3	2028	2044	16	2025	2036	11
IPSL-CM5A-LR	2015	2030	15	2012	2028	16
IPSL-CM5A-MR	2019	2031	12	2016	2032	16
IPSL-CM5B-LR	2027	2052	25	2024	2038	14
MIROC5	2040	2068	28	2035	2050	15
MIROC-ESM	2021	2034	13	2021	2030	9
MIROC-ESM-CHEM	2022	2037	15	2019	2029	10
MPI-ESM-LR	2025	2040	15	2015	2036	21
MPI-ESM-MR	2024	2045	21	2018	2040	22
MRI-CGCM3	2050	2084	34	2041	2053	12
NorESM1-M	2039	2074	35	2033	2049	16
mean			19			13

3. Ideally, it would have been better to use a lower level emission scenario (RCP2.6 for instance).

Response: Thank you for your suggestion. Lower emission scenarios like RCP2.6 and RCP4.5 are more consistent with the low warming target. However, among the CMIP5 models that have provided daily precipitation data in the archive, only 10 models would reach a warming of 2°C under RCP2.6 before 2100. An ensemble of 10 models may be insufficient to derive reliable confidence intervals, in particular to remove the impact

of internal decadal variability.

In the revised manuscript, we have analyzed the results of RCP4.5 (Please see Supplementary Figs. S5-S6). The results support our conclusions derived from RCP8.5, in: (1) nonlinear increases in exposure to ‘dangerous’ extremes with higher warming; (2) robustly avoided impacts by the 1.5°C warming compared with 2°C. Although the reduced exposure by the 0.5°C warming is slightly lower under RCP4.5 than RCP8.5, the difference lies within the range of inter-model spread. Thus, the robustness of these results adds fidelity to our conclusion.

The results of the emission scenario RCP4.5 are presented and noted in the revised manuscript (Please see L175-179 and Supplementary Figs. S5-S6).

4. Also, as the paper is clearly about the differences between 1.5 and 2°C warming in the monsoon regions, any discussion about other changes in temperatures or other regions should be removed to make the communication more clear.

Response: Accepted. The revised manuscript focuses on extreme precipitation in the monsoon regions.

5. My other main concern is for the precipitation above 95th percentile. Does it include tropical cyclones too? It could be better to separate the contribution from TC and monsoon itself for region with high TC impact, or at least have some discussion on this

point, as most of the CMIP5 models may not well represent the TC.

Response: Thank you for your suggestions. TCs do contribute to extreme precipitation in CMIP5 simulations (Utsumi et al., 2016 & their Figure 7). In climatology, TCs contribute typically less than 10% to accumulated precipitation above the 99th and 99.9th percentile in the East Africa, East Asia, and Central America, and 30-40% for Southeast Asia. In future projections, TCs contribute less than 20% of the increase in precipitation above the 99th and 99.9th percentile for the above regions. The contribution from TCs increases with extreme levels. Other systems like monsoon and extratropical cyclones dominate the changes in extreme precipitation in CMIP5 simulations (>80% of contribution).

As you noted, the TC activity is not well simulated by the CMIP5 models. Specifically, the global TC frequency is highly underestimated by CMIP5 models compared with observations. There is also a significant deficiency in the geographical patterns of TC tracks and formation. Factors hindering the simulation of TC activity includes horizontal resolution and SST bias (Camargo 2013). As suggested, considering the limited ability of CMIP5 models in simulating TCs and large uncertainties in its projections (Camargo 2013; Emanuel et al., 2013), we do not distinguish the contributions from TCs and monsoons to changes in extreme rainfall. The TC-related extreme precipitation issue deserves further investigation by employing high-resolution models. We include the following discussion in the revised manuscript (Please see L78-86):

The weaker response of the RX5day in simulations compared to the thermodynamic arguments implies a potential offset from the dynamic changes (blue vs. red curves in Fig. 2a). Changes in dynamic circulation can substantially affect extreme precipitation¹⁹⁻²¹. The underlying mechanisms include the weakening of the large-scale monsoon circulation^{11, 22-25}, which results from the stabilization of the tropical atmosphere associated with global warming²⁶⁻²⁷, the modulation of regional monsoon circulation by land-sea thermal contrast changes²⁸⁻²⁹, the gradient of sea surface temperature warming patterns³⁰, and changes in synoptic scale circulations, such as monsoon depressions^{21, 29, 31} and tropical cyclones³².

6. Finally, I think it could be worth it to improve the comparison between 1.5 and 2°C periods by including also a comparison with a linear interpolation from 1.5°C impact to 2°C (basically, just multiply the results from 1.5°C by a factor of [2/1.5], and compare them with the actual modelled 2°C warming). This would be a good way to discuss if the response in the extremes and vulnerability is purely linear to the temperatures or not.

Response: Thank you for your suggestion. Whether precipitation extremes respond linearly to temperature increase is both important and interesting. Based on your suggestion, we have conducted two aspects of analysis:

(1) The response of the mean state of extremes

We examined the changes of extreme precipitation versus global mean surface air

temperature (Supplementary Fig. S3a). For RX5day averaged over the GM region, the responses to temperature are nearly linear in individual models, with a multimodel median scaling rate of 5.17%/K under RCP8.5. Such linear responses of the mean state are generally expected from the thermodynamic arguments, where changes in extreme precipitation are largely determined by moistening of the atmosphere by around 7%/K (Trenberth, 1999; Trenberth et al., 2003; Chou et al., 2011). The weaker response in comparison to the theoretical estimation based on the Clausius-Clapeyron equation implies the offset by dynamical processes.

(2) The response of high-ranking extreme events

The high-ranking events, e.g., the ‘dangerous’ events exceeding 1-4 σ defined in this study, lie in the upper tail of the PDF. Both the rightward shift and broadening of PDFs with higher warming levels lead to nonlinear increases in the upper tail extremes (Fig. 4a in the revised manuscript). We quantitatively compare the probability of 1-4 σ exceedance events at different warming worlds ranging from the 0°C to a 4°C warming in the revised Fig. 4c. At warming levels above 2°C, the area of exposure exceeds that linearly extrapolated from those in 0°C and 1.5°C conditions. The population exposure also exhibits nonlinear increases although population distribution also plays a role here.

In summary, (1) the mean state of extreme precipitation responds approximately linearly to global warming; (2) the probability of high-ranking extreme events increases beyond linear expectation at warmer levels, and thus for both area and population exposures.

We noted the linear response of extreme precipitation to global warming in terms of the mean state and highlighted the nonlinear increases in exposure to ‘dangerous’ events in the revised manuscript (Please see L71-73 and L165-169).

References:

Camargo, S. J. (2013). Global and regional aspects of tropical cyclone activity in the cmip5 models. *Journal of Climate*, 26(24), 9880-9902.

Chou, C., Chen, C. A., Tan, P. H., & Chen, K. T. (2011). Mechanisms for global warming impacts on precipitation frequency and intensity. *Journal of Climate*, 25(9), 3291-3306.

Emanuel, K. A. (2013). Downscaling CMIP5 climate models shows increased tropical cyclone activity over the 21st century. *Proceedings of the National Academy of Sciences of the United States of America*, 110(30), 12219.

Han, Z., Luo, F., Shuanglin, L. I., Gao, Y., Furevik, T., & Svendsen, L. (2016). Simulation by CMIP5 models of the Atlantic Multidecadal Oscillation and its climate impacts. *Advances in Atmospheric Sciences*, 33(12), 1329-1342.

Trenberth, K. E. (1999). Conceptual framework for changes of extremes of the hydrological cycle with climate change. *Climatic Change*, 42(1), 327-339.

Trenberth, K. E., Dai, A., Rasmussen, R. M., & Parsons, D. B. (2003). The changing character of precipitation. *Bulletin of the American Meteorological Society*, 84(9), 1205-1217.

Utsumi, N., Kim, H., Kanae, S., & Oki, T. (2016). Which weather systems are projected to cause future changes in mean and extreme precipitation in CMIP5 simulations?. *Journal of Geophysical Research*, 121(18).

Reviewers' comments:

Reviewer #1 (Remarks to the Author):

Second Review for Zhang et al (2018) for Nature Communications.

Summary:

The authors present a summary of changes in extreme precipitation and population exposure under 1.5°C and 2°C of warming from global climate models and population scenarios. Results are not unexpected, but fit within the 1.5°C vs. 2°C debate.

The authors have responded in detail to the points raised by the reviewers upon the earlier manuscript. I appreciate the authors' efforts for their additional, detailed analysis.

Through these changes, the manuscript has gained focus.

(I had been Reviewer #1 in the previous round of review).

However, unfortunately, there are still a number of severe technical problems in the paper (severely affecting the main results), and issues related to inaccurate presentation and discussion, which I outline below. In summary, the most severe problem is that the choice of 4-sigma exceedances of an extreme index as a metric to quantify extreme precipitation across space leads to artifacts (and apparently very large avoided impacts for 1.5°C as compared to 2°C of warming for these 4-sigma exceedances). This is because any reasonable statistical inference of a one-in-15787-years extreme event

(i.e. 4-sigma exceedance) is simply not possible from 56 data points (i.e. a 56 year reference period).

Major issues:

* Technical correctness of key results: Choice of metric and reference period *

The authors have chosen a different metric for extreme precipitation in the revised manuscript. This metric is based on a "normalization procedure of extreme indices" (p. 13, l. 254). Although not explicitly stated, I assume that the authors mean that Rx5day means have been subtracted and divided by the standard deviation, both

estimated from a 56-year long base period. Then, the main results of the paper, as stated e.g. in the Abstract and Conclusion, are presented in "sigma-exceedances" (specifically, in exceeding 4 sigma). Although the authors are aware of this problem (judging from their rebuttal letter), this issue still presents several severe problems:

1. Normalization and interpretation of sigma exceedances is appropriate only if the data are approximately Gaussian. This is certainly not the case here, as extreme precipitation, i.e. block maxima, are well known to follow a (skewed) extreme value distribution.

2. The authors choose 4-sigma exceedances, which leads to very large "avoided impacts" (as compared to, say, avoided impacts for a 1- or 2-sigma exceedance).

However, even if the data were Gaussian, a 4-sigma extreme would occur -on average- every 15787 years; and any reasonable

statistical inference for an extreme of this kind is just not possible from 56 data points in the reference period.

For example, comparing the reference period with out-of-base period for some arbitrary GEV distribution using 4-sigma exceedances can lead to a 2.5-fold inflation of 4-sigma extremes (see sample R-Code below).

3. Finally, the authors communicate the most extreme "avoided impacts" of e.g. 118% in the Abstract, rather than more modest estimates that they would encounter if a more reasonable exceedance threshold would be used.

That being said, I appreciate that the authors have taken the previous critique seriously and that they have tested a number of different thresholds and options. In particular, using extreme value

statistics to estimate 10-year and 20-year return periods, as the authors have done in Supplementary Figure 3 seems a reasonable choice and would overcome the points 1-3 above.

Sample code to show how normalisation changes occurrence of extremes:

```
install.packages("extRemes")
library(extRemes)

extremes.base.period = sapply(1:10000, FUN=function(x) revd(n = 56, loc = 0, scale = 1,
type="GEV"))
extremes.base.period.mean = apply(X = extremes.base.period, MARGIN = 2, FUN = function(x)
mean(x))
extremes.base.period.sd = apply(X = extremes.base.period, MARGIN = 2, FUN = function(x)
sd(x))
extremes.base.period.norm = apply(X = extremes.base.period, MARGIN=2, FUN=function(x) (x -
mean(x)) / sd(x))
extremes.future.period.norm = sapply(1:10000, FUN=function(ind) (revd(n = 20, loc = 0, scale =
1, type="GEV") - extremes.base.period.mean[ind]) / extremes.base.period.sd[ind])

length(which(c(extremes.base.period.norm[37:56,]) > 4)) # Number of extremes in 20 years of
reference period
length(which(c(extremes.future.period.norm > 4))) # Number of extremes outside the reference
period in the absence of trend in extremes.
```

* Discussion, Interpretation, and Presentation of Results *

Some of the discussion and interpretation of results, in particular in the Abstract and Conclusions, seems inaccurate

and overstated given the presented evidence.

This leads to key results in the Abstract (and emphasized throughout the paper) that read for instance

(a) "... avoided impacts by 0.5°C less warming

amount to 118% ..." (l. 21-22), and (b) "Nonlinear increases in exposure with further warming highlight the importance

and necessity of realizing the 1.5°C warmer world" (l. 28-30). However, both statements are inaccurate given the presented

evidence, because:

(a) 118% refer to 4-sigma exceedances, which cannot be reliably estimated (see comment above).

If results for lower threshold

exceedances (but more practically important ones) would be given, for instance w.r.t. a 10-year or 20-year return period

(as presented in Supp. Figure 3), "avoided impacts" would be at approximately 8-12%, rather than 118%. This is a 10-fold (!) difference, and shows how strongly "avoided impacts" depend on the chosen metric (i.e. the very high values, 118% etc., occur only because these events are almost absent at present, which should also be clarified in the Abstract).

(b) The authors emphasize many times the importance of "nonlinear increases in exposure with further warming" (e.g. l. 29-30 in the Abstract and throughout the manuscript). The evidence for this statement appears to stem from Figure 4c.

However, these statements are confusing at the very least, because the deviation of the linear fit emerges in most cases

only at very high warming levels (3°C and higher), rather than at 1.5 or 2°C as might be inferred from the Abstract or

title of the paper. Moreover, if the linear fits would have excluded the pre-industrial period (in which extended

extreme precipitation events were largely absent), then the increases in exposure might follow a linear line actually quite well.

* Thermodynamic vs. dynamic changes and theoretical increase in variability of extreme precipitation *

The authors state in l. 99/100: "This results in a theoretical increase in the variability by 7%/K": Why do you expect a theoretical increase in the variability by 7%/K due to the Clausius Clapeyron relationship? Clearly, increases in precipitation extremes are expected, and the C-C relationship gives a good physical argument, which can reasonably be extended to argue that the variability of extreme value distributions is increasing. But why should this increase in variability follow 7%/K? It would only do so, if each precipitation event in the distribution (i.e. the whole distribution) would increase by 7%/K.

Even if we had a good theoretical idea of changes in the upper tail of the (extreme) precipitation distribution, how should we be able to derive theoretical changes in the variability from this, as we do not know what is happening in the

lower tail? This means the assumed 7%/K increase in variability (i.e. SD) would only hold if the 7%/K increase would also hold for years with low Rx5day.

However, there has been some recent work on the topic (e.g. Pendergrass et al., 2017, Sci. Rep), which also briefly describes a number of plausible hypotheses regarding changes in precipitation variability (but not extreme precipitation variability).

Changes in variability in extreme precipitation are an interesting aspect of the paper, but it would be necessary that the authors explain where their "theoretical expectation for changes in variability" comes from.

In addition, the paragraph in l. 101-109 that discusses this topic is hard to understand and I would encourage rephrasing.

l.80-86: Thermodynamic vs. dynamic changes.

The authors give a number of plausible reasons why the large-scale monsoon circulation might be weakening, as an explanation for the relatively smaller increase in simulated Rx5day as compared to the theoretical argument in Fig. 2. However, I am not sure whether these arguments do reflect the data of Fig. 2b appropriately: This is because changes in precipitable water extremes seem to follow C-C quite well. If monsoon circulation would indeed weaken significantly, the amount of Precipitable water should also decrease, wouldn't it?

Minor comments:

p. 10, l. 190-191: Does the fact that models were selected on the basis that the difference in the timings

of 1.5°C and 2°C warming is no less than 9 years systematically exclude some models that maybe produce this transition relatively quickly? If so, this would potentially produce a systematic bias, and thus should be checked.

Fig 3d,e,f: Although I acknowledge that measures such as RX5day X population count are used in the literature, and the authors have replied to this concern already in the previous round of review, personally I am not convinced by the practical usefulness and relevance of these kind of "impact" measures: In terms of the "impacts" diagnosed by this measure, a doubling of the magnitude of RX5day with population hold constant would have the same impact as a doubling of the population count or density while holding RX5day constant. Hence, this measure assumes complete linearity and inter-changeability between changes in RX5day and population count. While

the authors emphasize "non-linear changes with warming" throughout the manuscript, this kind of non-linearity in impacts seem to be disregarded, and should be at least mentioned if publication in a broad readership journal is anticipated.

Reviewer #2 (Remarks to the Author):

The authors have addressed all my comments and the manuscript may now be accepted.

Reviewer #3 (Remarks to the Author):

The authors have take time to address every comment in a good way and to modify the manuscript accordingly to the suggestion. So in my opinion the study is ready for publication in Nature Communication.

Reply to Reviewer #1:

Thank you for the insightful comments and detailed instruction on how to improve the manuscript. The quality of the manuscript has been greatly improved based on your comments. In the following, the texts with *italic font* are your original comments, and the texts with normal font are our responses.

Reviewer #1 (Remarks to the Author)**Summary:**

The authors present a summary of changes in extreme precipitation and population exposure under 1.5°C and 2°C of warming from global climate models and population scenarios. Results are not unexpected, but fit within the 1.5°C vs. 2°C debate. The authors have responded in detail to the points raised by the reviewers upon the earlier manuscript. I appreciate the authors' efforts for their additional, detailed analysis. Through these changes, the manuscript has gained focus. (I had been Reviewer #1 in the previous round of review). However, unfortunately, there are still a number of severe technical problems in the paper (severely affecting the main results), and issues related to inaccurate presentation and discussion, which I outline below. In summary, the most severe problem is that the choice of 4-sigma exceedances of an extreme index as a metric to quantify extreme precipitation across space leads to artifacts (and apparently very large avoided impacts for 1.5°C as compared to 2°C of warming for these 4-sigma exceedances). This is because any reasonable statistical inference of a one-in-15787-years extreme event (i.e. 4-sigma exceedance) is simply not possible from 56 data points (i.e. a 56 year reference period).

Response: Thank you for the insightful comments and suggestions. We agree that your concern on the use of σ -exceedance metric based on normalization (especially the high threshold of 4σ) to quantify the different impacts of extreme precipitation in warming scenarios is reasonable. Our major revision of the manuscript is that we now employ the 10-year and 20-year return values from the baseline as thresholds for ‘dangerous’ extremes in the text as you recommended. The revision guided by your comments, in particular the choice of the more reasonable metric has enhanced the robustness of the conclusions. Please see our detailed responses below.

Major issues:

1. Technical correctness of key results: Choice of metric and reference period

The authors have chosen a different metric for extreme precipitation in the revised manuscript. This metric is based on a “normalization procedure of extreme indices” (p. 13, l. 254). Although not explicitly stated, I assume that the authors mean that R_{x5day} means have been subtracted and divided by the standard deviation, both estimated from a 56-year long base period. Then, the main results of the paper, as stated e.g. in the Abstract and Conclusion, are presented in “sigma-exceedances” (specifically, in exceeding 4 sigma). Although the authors are aware of this problem (judging from their rebuttal letter), this issue still presents several severe problems:

1). Normalization and interpretation of sigma exceedances is appropriate only if the data are approximately Gaussian. This is certainly not the case here, as extreme precipitation, i.e. block maxima, are well known to follow a (skewed) extreme value

distribution.

2). *The authors choose 4-sigma exceedances, which leads to very large “avoided impacts” (as compared to, say, avoided impacts for a 1- or 2-sigma exceedance). However, even if the data were Gaussian, a 4-sigma extreme would occur -on average- every 15787 years; and any reasonable statistical inference for an extreme of this kind is just not possible from 56 data points in the reference period. For example, comparing the reference period with out-of-base period for some arbitrary GEV distribution using 4-sigma exceedances can lead to a 2.5-fold inflation of 4-sigma extremes (see sample R-Code below).*

3). *Finally, the authors communicate the most extreme “avoided impacts” of e.g. 118% in the Abstract, rather than more modest estimates that they would encounter if a more reasonable exceedance threshold would be used.*

That being said, I appreciate that the authors have taken the previous critique seriously and that they have tested a number of different thresholds and options. In particular, using extreme value statistics to estimate 10-year and 20-year return periods, as the authors have done in Supplementary Figure 3 seems a reasonable choice and would overcome the points 1-3 above.

Response: Thank you for the insightful comment. In our previous manuscript, we analyzed extreme events exceeding 1-4 σ (inter-annual standard deviation in the baseline), to represent different levels of ‘dangerous’. As you mentioned, normalization was performed to the original time series of RX5day, where the means have been subtracted and divided by the standard deviation, both estimated from the 1950-2005 base period. We agree that the previously used 3 σ - or 4 σ -exceedances are

too high.

As you recommended, we now employ the 10-year and 20-year return values from the 1950-2005 baseline as thresholds for ‘dangerous’ extremes in the revision, which is supposed to be a reasonable choice and would overcome the three issues listed above. The results based on the new metric are shown in Figs. 4-5 in the revised manuscript. The more reasonable thresholds have led to reasonable “avoided impacts” in the revised manuscript (please see L20-23 of the revised manuscript).

It is interesting to note that a comparison of the results based on the new return value thresholds (Figs. 4-5 in the revised manuscript) with those based on 1-2 σ exceedances (Supplementary Figs. S6-S7) shows comparable results. Specifically, the thresholds of 1.30 σ and 1.65 σ correspond to the 10-year and 20-year return values for the Gaussian distribution, respectively (see Table R1 below). The areal and population exposures to ‘dangerous’ extremes, as well as the avoided impacts by the half a degree less warming are quantitatively comparable between the two metrics, confirming the robustness of our conclusion (Tables R2-R3 below).

Thus, we employ the 10-year and 20-year return values as the metric to define ‘dangerous’ extremes in the revised main manuscript; meanwhile, we show the results based on 1-2 σ exceedances in the Supplementary Information (SI, Figs. S6-S7) as additional information. We think this additional information is useful and thus present it as a discussion in the Methods section. Whether we should keep it as this or remove it from both the manuscript and the SI, you are appreciated to give suggestions.

Please see L180-184 and L282-301 of the revised manuscript.

Table R1. Cumulative probability and return period (in year) for different σ

thresholds for a Gaussian distribution.

σ threshold	Cumulative probability	Return period (year)
1.00 σ	0.8413	6.30
1.30 σ	0.9032	10.33
1.65 σ	0.9505	20.20
2.00 σ	0.9772	43.86

Table R2. Areal and population (the GPW2000 case) exposures to ‘dangerous’ extremes reduced by the 1.5°C warming compared with 2°C over the GM region. The ‘dangerous’ extremes are defined as those exceeding the baseline 10-year and 20-year return values (RVs). The multimodel medians and interquartile ranges are shown.

RV exceedance	Land area	GPW2000
10-yr RV	19.5% (13.4-30.7%)	26.8% (13.3-35.6%)
20-yr RV	24.9% (18.0-40.6%)	36.5% (21.8-46.1%)

Table R3. Same as Table R2 except that the ‘dangerous’ extremes are defined as those exceeding 1.00 σ , 1.30 σ , 1.65 σ , and 2.00 σ , based on normalization.

σ exceedance	Land area	GPW2000
1.00 σ	14.5% (11.4-23.6%)	21.3% (9.7-26.7%)
1.30 σ	19.6% (15.0-29.2%)	25.5% (18.3-33.0%)
1.65 σ	22.1% (13.3-37.9%)	33.3% (20.8-41.0%)
2.00 σ	30.9% (16.3-41.9%)	34.6% (25.2-47.3%)

* 1.30σ and 1.65σ correspond to the 10-year and 20-year return values for the Gaussian distribution.

2. Discussion, Interpretation, and Presentation of Results

Some of the discussion and interpretation of results, in particular in the Abstract and Conclusions, seems inaccurate and overstated given the presented evidence. This leads to key results in the Abstract (and emphasized throughout the paper) that read for instance

(a) “...avoided impacts by 0.5°C less warming amount to 118% ...” (l. 21-22), and (b) “Nonlinear increases in exposure with further warming highlight the importance and necessity of realizing the 1.5°C warmer world” (l. 28-30). However, both statements are inaccurate given the presented evidence, because:

(a) 118% refer to 4-sigma exceedances, which cannot be reliably estimated (see comment above). If results for lower threshold exceedances (but more practically important ones) would be given, for instance w.r.t. a 10-year or 20-year return period (as presented in Supp. Figure 3), “avoided impacts” would be at approximately 8-12%, rather than 118%. This is a 10-fold (!) difference, and shows how strongly “avoided impacts” depend on the chosen metric (i.e. the very high values, 118% etc., occur only because these events are almost absent at present, which should also be clarified in the Abstract).

Response: Thank you for the comment. Yes, employing 3- and 4- σ exceedances in the previous manuscript overstated the avoided impacts. Thus, as suggested in your major

comment #1, we now employ the more reasonable *10-year and 20-year return values* from the baseline as thresholds for ‘dangerous’ extremes in the revision. The “avoided impacts” have been recalculated correspondingly. A more reasonable estimate is now presented, i.e., avoided impacts of 25% (18%-41%) and 36% (22%-46%), respectively, for areal and population exposures to RX5day events exceeding the baseline 20-year return values over the GM region. Please see L20-23 of the revised manuscript.

In addition, the avoided impacts for 10-year or 20-year return value metrics should be 19% (13%-31%) and 25% (18%-41%) for areal exposure rather than 8-12% as noted in our previous version. The previous estimation was biased due to a misunderstanding of the sign of the shape parameter in the NCL function (“extval_mlegev”) which was used to estimate the parameters for the GEV distribution. We confirm the correctness of our current NCL codes by reproducing the figures of Kharin et al. (2013) (please see the Appendix of this response letter).

(b) The authors emphasize many times the importance of “nonlinear increases in exposure with further warming” (e.g. l. 29-30 in the Abstract and throughout the manuscript). The evidence for this statement appears to stem from Figure 4c. However, these statements are confusing at the very least, because the deviation of the linear fit emerges in most cases only at very high warming levels (3°C and higher), rather than at 1.5 or 2°C as might be inferred from the Abstract or title of the paper. Moreover, if the linear fits would have excluded the pre-industrial period (in which extended extreme precipitation events were largely absent), then the increases in exposure might follow a linear line actually quite well.

Response: Thank you for the comment. We agree that the deviation of the linear fit emerges in most cases only at very high warming levels (3°C and higher), rather than at 1.5°C or 2°C which is the focus of this study (please see Fig. 4a in the revised manuscript). Thus, the statement in the text is now revised as “Whereas the increase in exposure is quasi-linear below the 2°C warming level, nonlinear increases emerge at higher warming levels (higher than 2°C), implying that excessively amplified impacts from ‘dangerous’ extremes could be induced”. The original statements in the abstract and summary are deleted accordingly. Please see L134-137 of the revised manuscript.

3. Thermodynamic vs. dynamic changes and theoretical increase in variability of extreme precipitation

1) The authors state in l. 99/100: “This results in a theoretical increase in the variability by 7%/K”.

Why do you expect a theoretical increase in the variability by 7%/K due to the Clausius Clapeyron relationship? Clearly, increases in precipitation extremes are expected, and the C-C relationship gives a good physical argument, which can reasonably be extended to argue that the variability of extreme value distributions is increasing. But why should this increase in variability follow 7%/K? It would only do so, if each precipitation event in the distribution (i.e. the whole distribution) would increase by 7%/K.

Even if we had a good theoretical idea of changes in the upper tail of the (extreme) precipitation distribution, how should we be able to derive theoretical changes in the variability from this, as we do not know what is happening in the lower tail? This means the assumed 7%/K increase in variability (i.e. SD) would only hold if the 7%/K increase would also hold for years with low Rx5day.

However, there has been some recent work on the topic (e.g. Pendergrass et al., 2017, Sci. Rep), which also briefly describes a number of plausible hypotheses regarding changes in precipitation variability (but not extreme precipitation variability).

Changes in variability in extreme precipitation are an interesting aspect of the paper, but it would be necessary that the authors explain where their “theoretical expectation for changes in variability” comes from. In addition, the paragraph in l. 101-109 that discusses this topic is hard to understand and I would encourage rephrasing.

Response: Thank you for your comments. In the previous version of manuscript, we discussed the changing variability of extreme precipitation because we had used the σ -exceedance metric, which was based on normalization. We applied an idealized thermodynamic assumption based on the C-C relationship, intending to estimate the thermodynamic contribution to changes in extreme precipitation, as in previous studies (Allan and Soden, 2008; Brown et al., 2017). In this idealized assumption, each RX5day event was assumed to intensify by 7%/K, resulting in an assumed increase in variability also by 7%/K. This is only an idealized assumption, we agree that it should not be hoped that each RX5day event in the distribution (i.e. the whole distribution) would increase by 7%/K.

The increase in extreme precipitation is dominated by an increase in atmospheric moisture while partly offset by changing vertical circulation via vertical moisture advection in the monsoon region (Freychet et al., 2015). The increasing variability in extreme precipitation has also been reported under global warming in model projections (Kharin and Zwiers, 2005). Some recent work studying changes in mean precipitation variability has proposed plausible hypotheses that increases in moisture (i.e., thermodynamic responses) provide a first-order explanation for increasing precipitation variability, which is also modulated by circulation (Brown et al., 2017; Pendergrass et al., 2017).

As suggested in your major comment #1, we now mainly use return values as thresholds in the revised manuscript, the normalization by the standard deviation (i.e., variability) is no longer needed in the text. More importantly, the objective of this study is to estimate the avoided impact of extreme precipitation by the 0.5°C less warming, while mechanism for changing variability in extreme precipitation is not the focus of the paper. Thus, to keep the topic of the paper being focused on our motivation, we only show the model simulated increase in variability of extreme precipitation in the Supplementary Information as background information, noting that this model response is consistent with previous research (Kharin and Zwiers, 2005). The comparison of model response to that estimated from the idealized C-C relationship assumption is no longer needed. Please see L104-107 of the revised manuscript and Supplementary Fig. S3.

2) L80-86: Thermodynamic vs. dynamic changes.

The authors give a number of plausible reasons why the large-scale monsoon

circulation might be weakening, as an explanation for the relatively smaller increase in simulated Rx5day as compared to the theoretical argument in Fig. 2. However, I am not sure whether these arguments do reflect the data of Fig. 2b appropriately: This is because changes in precipitable water extremes seem to follow C-C quite well. If monsoon circulation would indeed weaken significantly, the amount of precipitable water should also decrease, wouldn't it?

Response: Thank you for the question. The model projected weakening monsoon circulation along with increasing precipitable water is not contradictory. The weakening of large-scale monsoon circulation results from the stabilization of the tropical atmosphere associated with global warming (Held and Soden, 2006; Vecchi and Soden, 2007), and modulated by land-sea thermal contrast changes (Ueda et al., 2006; Turner and Annamalai, 2012), while the increase in precipitable water is related to increasing water holding capacity of the warmer atmosphere. The weakening of monsoon circulation while increasing atmospheric moisture have been well established in future projections (e.g., Hsu et al., 2012, 2013; Kitoh et al., 2013; Chen and Zhou, 2015) and has been summarized in IPCC WG1 AR5 (Christensen et al., 2013).

Minor comments:

1. p. 10, l. 190-191: Does the fact that models were selected on the basis that the difference in the timings of 1.5°C and 2°C warming is no less than 9 years systematically exclude some models that maybe produce this transition relatively

quickly? If so, this would potentially produce a systematic bias, and thus should be checked.

Response: Thank you for your suggestion. Among the CMIP5 models, 23 of them meet the criteria of (1) daily precipitation data availability and (2) reaching a 2°C warming before 2100 under RCP4.5 and RCP8.5 scenarios. Among them, two models produce a transition from the 1.5°C to 2°C in less than 9 years, i.e., FGOALS-s2 and CSIRO-Mk3-6-0 (see Table R4 below). Thus, the remaining 21 models are used in the manuscript.

As suggested, we compare the key results from the selected 21 models and all 23 models (cf. Figs. 4-5 in the revised manuscript and Figs. R1-R2 below). Here we do not show the results from the two models (FGOALS-s2 and CSIRO-Mk3-6-0) specifically, because the difference between the 1.5°C and 2°C warming levels in a single realization of a model is obscured by climate internal variability. The comparison shows quantitatively comparable results between the two sets of models (see Table R5 below). Thus excluding the two models does not affect the conclusion. We have noted this in the Methods section in the revised manuscript (please see L202-205).

Table R4. The timings (in year) of 1.5°C and 2°C warming and their differences under RCP4.5 and RCP8.5 emission scenarios, respectively.

	RCP4.5			RCP8.5		
Model	1.5°C	2°C	ΔT	1.5°C	2°C	ΔT

ACCESS1-0	2030	2052	22	2025	2042	17
ACCESS1-3	2038	2055	17	2031	2042	11
bcc-csm1-1	2022	2043	21	2021	2036	15
bcc-csm1-1-m	2015	2035	20	2012	2028	16
CanESM2	2018	2032	14	2013	2027	14
CCSM4	2017	2040	23	2015	2029	14
CESM1-BGC	2021	2044	23	2017	2034	17
CMCC-CM	2035	2052	17	2030	2040	10
CMCC-CMS	2033	2053	20	2029	2042	13
CNRM-CM5	2035	2056	21	2029	2043	14
CSIRO-Mk3-6-0	2035	2049	14	2035	2043	8
FGOALS-s2	2010	2012	2	2010	2012	2
GFDL-CM3	2028	2044	16	2025	2036	11
IPSL-CM5A-LR	2015	2030	15	2012	2028	16
IPSL-CM5A-MR	2019	2031	12	2016	2032	16
IPSL-CM5B-LR	2027	2052	25	2024	2038	14
MIROC5	2040	2068	28	2035	2050	15
MIROC-ESM	2021	2034	13	2021	2030	9
MIROC-ESM- CHEM	2022	2037	15	2019	2029	10
MPI-ESM-LR	2025	2040	15	2015	2036	21
MPI-ESM-MR	2024	2045	21	2018	2040	22
MRI-CGCM3	2050	2084	34	2041	2053	12
NorESM1-M	2039	2074	35	2033	2049	16

Figure R1. Same as Figure 4 in the revised manuscript, but derived from all the 23 models under the RCP8.5 projections.

Figure R2. Same as Figure 5 in the revised manuscript, but derived from all the 23 models under the RCP8.5 projections.

Table R5. Avoided impacts, i.e., areal and population exposures to ‘dangerous’ extremes reduced by the 1.5°C warming compared to 2°C, over the GM region. The ‘dangerous’ extremes are defined as those exceeding the baseline 10-year and 20-year return values (RVs). The multimodel medians and interquartile ranges are shown.

	RV exceedance	Land area	GPW2000
21 models	10-yr RV	19.5% (13.4-30.7%)	26.8% (13.3-35.6%)
	20-yr RV	24.9% (18.0-40.6%)	36.5% (21.8-46.1%)
23 models	10-yr RV	18.5% (11.0-28.3%)	21.5% (8.6-29.9%)
	20-yr RV	24.5% (14.6-39.4%)	33.4% (18.9-43.4%)

2. Fig 3d,e,f: Although I acknowledge that measures such as RX5day X population count are used in the literature, and the authors have replied to this concern already in the previous round of review, personally I am not convinced by the practical usefulness and relevance of these kind of “impact” measures: In terms of the “impact” diagnosed by this measure, a doubling of the magnitude of RX5day with population hold constant would have the same impact as a doubling of the population count or density while holding RX5day constant. Hence, this measure assumes complete linearity and inter-changeability between changes in RX5day and population count. While the authors emphasize “non-linear changes with warming” throughout the manuscript, this kind of non-linearity in impacts seem to be disregarded, and should be at least mentioned if publication in a broad readership journal is anticipated.

Response: Thank you for the comments. We agree that the measure of RX5day X population is not a rigorous measure of “impacts”, as it assumes complete linearity and inter-changeability between changes in RX5day and population count. Thus, in the revised manuscript, we no longer use this measure. Instead, we interpret the changes in RX5day induced by the half a degree additional warming as:

“Robust increases in RX5day are projected to mostly affect the Asian and African monsoon regions with the half a degree additional warming due to the large sensitivity of extreme precipitation to global warming in these regions (Fig. 3c). Note

these regions have dense populations (Fig. 1a”).

Please see L96-99 of the revised manuscript.

Appendix

We reproduced Supplementary Fig. S6 in Kharin et al. (2013), showing the return values of annual maximum daily precipitation (Rx1day) from three datasets, which verifies our calculation.

Figure R3. 20-year return values of 1986-2005 annual maximum daily precipitation (Rx1day) in CMIP5 multimodel ensemble median (upper), ERA-Interim (middle), and NCEP2 (lower) (unit: mm). The CMIP5 models used here (i.e., used in our study) are slightly different from those used in Kharin et al. (2013).

References

- Allan, R. P., & Soden, B. J. Atmospheric warming and the amplification of precipitation extremes. *Science*, **321**, 1481 (2008).
- Brown, J. R., Moise, A. F., & Colman, R. A. Projected Increases in Daily to Decadal Variability of Asian - Australian Monsoon Rainfall. *Geophys. Res. Lett.* (2017).
- Chen, X., and T. Zhou. Distinct effects of global mean warming and regional sea surface warming pattern on projected uncertainty in the South Asian summer monsoon. *Geophys. Res. Lett.* **42** (2015).
- Christensen, J.H., et al. Climate Phenomena and their Relevance for Future Regional Climate Change. In: *Climate Change 2013: The Physical Science Basis. Contribution of Working Group I to the Fifth Assessment Report of the Intergovernmental Panel on Climate Change*. Cambridge University Press, Cambridge, United Kingdom and New York, NY, USA. (2013).
- Freychet, N., Hsu, H. H., Chou, C. & Wu, C. H. Asian summer monsoon in CMIP5 projections: a link between the change in extreme precipitation and monsoon dynamics. *J. Clim.* **28**, 1477-1493 (2015).
- Held, I. M., & Soden, B. J. Robust responses of the hydrological cycle to global warming. *J. Clim.* **19**, 5686-5699 (2006).
- Hsu, P. C., et al. Increase of global monsoon area and precipitation under global warming: A robust signal? *Geophys. Res. Lett.*, **39** (2012).
- Hsu, P. C., Li, T., Murakami, H., & Kitoh, A. Future change of the global monsoon revealed from 19 CMIP5 models. *J. Geophys. Res.* **118**, 1247-1260 (2013).
- Kharin, V. V., & Zwiers, F. W. Estimating extremes in transient climate change

- simulations. *J. Clim.* **18**, 1156-1173 (2005).
- Kharin, V.V., Zwiers, F.W., Zhang, X. et al. Changes in temperature and precipitation extremes in the CMIP5 ensemble. *Climatic Change* **119** (2013).
- Kitoh, A. et al. Monsoons in a changing world: a regional perspective in a global context. *J. Geophys. Res.* **118**, 3053-3065 (2013).
- Pendergrass, A. G., Knutti, R., Lehner, F., Deser, C., & Sanderson, B. M. Precipitation variability increases in a warmer climate. *Sci. Rep.* **7** (2017).
- Turner, A. G., & Annamalai, H. Climate change and the south Asian summer monsoon. *Nat. Clim. Change* **2**, 587-595 (2012).
- Ueda, H., A. Iwai, K. Kuwako, & M. E. Hori. Impact of anthropogenic forcing on the Asian summer monsoon as simulated by eight GCMs, *Geophys. Res. Lett.*, **33**, L06703 (2006).
- Vecchi, G. A., & Soden, B. J. Global warming and the weakening of the tropical circulation. *J. Clim.* **20**, 4316-4340 (2007).

REVIEWERS' COMMENTS:

Reviewer #1 (Remarks to the Author):

The authors have responded in detail to all comments and concerns raised upon the earlier version of the manuscript; in particular previously strong over-interpretation of extremes far in the tails based on a short reference period have been removed. The paper's results are well-expected, but nonetheless might be useful and timely given the 1.5 vs. 2°C debate.

I think the paper could be accepted if the Editors would deem the research suitable for the journal Nature Communications.

Additional information for transparent peer review file

1. The data in the transparent peer review file is analyzed and figures are created with NCAR Command Language (NCL; <http://dx.doi.org/10.5065/D6WD3XH5>).

2. Citations to related datasets:

In **Figure R3 in the Response to Reviewer#1 file** in the 2nd round of revision, two reanalysis datasets, the European Centre for Medium-Range Weather Forecasts (ECMWF) Interim Reanalysis (ERA-Interim; Dee et al., 2011) and National Centers for Environmental Prediction / Department of Energy Reanalysis 2 (NCEP2; Kanamitsu et al., 2002), are used.

Data Availability:

ERA-Interim data can be acquired from <http://apps.ecmwf.int/datasets/>. NCEP2 data can be acquired from <https://www.esrl.noaa.gov/psd/data/gridded/data.ncep.reanalysis2.html>.

References

1. The NCAR Command Language (Version 6.4.0) [Software]. (2017). Boulder, Colorado: UCAR/NCAR/CISL/TDD. <http://dx.doi.org/10.5065/D6WD3XH5>.
2. Dee, D. P., et al. The ERA-Interim reanalysis: Configuration and performance of the data assimilation system. *Quarterly Journal of the royal meteorological society*, 2011, 137(656), 553-597.
3. Kanamitsu, M., et al. NCEP-DOE AMIP-II Reanalysis (R-2). *Bulletin of the American Meteorological Society*, 2002, 83(11), 1631-1643.